# Feeding *Spodoptera exigua* larvae with gut-derived *Escherichia sp.* increases larval juvenile hormone levels inhibiting cannibalism

Xing-Xing Du[1,2], Sheng-Kai Cao[1,2], Hua-Yan Xiao[1,2], Chang-Jin Yang[1,2], Ai-Ping Zeng[1,2], Gong Chen[1,2] & Huan Yu ![ORCID] [1,2 ✉]

Feed quality influences insect cannibalistic behavior and gut microbial communities. In the present study, *Spodoptera exigua* larvae were fed six different artificial diets, and one of these diets (Diet 3) delayed larval cannibalistic behavior and reduced the cannibalism ratio after ingestion. Diet 3-fed larvae had the highest gut bacterial load ($1.396 \pm 0.556 \times 10^{14}$ bacteria/mg gut), whereas Diet 2-fed larvae had the lowest gut bacterial load ($3.076 \pm 1.368 \times 10^{12}$ bacteria/mg gut). The gut bacterial composition and diversity of different diet-fed *S. exigua* larvae varied according to the 16S rRNA gene sequence analysis. Enterobacteriaceae was specific to the Diet 3-fed larval gut. Fifteen culturable bacterial isolates were obtained from the midgut of Diet 3-fed larvae. Of these, ten belonged to *Escherichia sp.* After administration with Diet 1- or 2-fed *S. exigua* larvae, two bacterial isolates (*SePC-12* and *-37*) delayed cannibalistic behavior in both tested larval groups. Diet 2-fed larvae had the lowest Juvenile hormone (JH) concentration and were more aggressive against intraspecific predation. However, *SePC-12* loading increased the JH hormone levels in Diet 2-fed larvae and inhibited their cannibalism. Bacteria in the larval midgut are involved in the stabilization of JH levels, thereby regulating host larval cannibalistic behavior.

[1] Hunan Provincial Key Laboratory for Biology and Control of Plant Diseases and Insect Pests, Hunan Agricultural University, Changsha 410128 Hunan, China.
[2] College of Plant Protection, Hunan Agricultural University, Changsha 410128 Hunan, China. ✉email: huanyu@hunau.edu.cn

Cannibalism or intraspecific predation is widespread in many insect species[1–3]. A typical case of cannibalism in insects is when female praying mantis prey on their mates to obtain nutrition for their developing offspring[4–6]. The prevalence of cannibalistic behavior in insect populations is often related to high population density, low food (host plant) quality, or extreme conditions, such as high temperature and humidity[1,7,8]. However, cannibalistic behavior has also been observed when food is not limiting[2]. Cannibalism increases the rate of development and removes potential competitors, thereby influencing insect population dynamics[9,10].

Among Lepidopteran insects, for instance, *Helicoverpa zea* (Boddie), cannibalism significantly impacts the larval population dynamics, causing up to 75% larval mortality[11]. *Spodoptera frugiperda* (Smith) larvae display cannibalism to obtain energy, proteins, and amino acids by consuming conspecifics[12]. He et al. pointed out that cannibalism in *S. frugiperda* is essential for the larvae to survive starvation or to successfully colonize new, nutrient-poor food plants because the winners in cannibalism switch to gluconeogenesis and the utilization of amino acids[13]. In addition, cannibalism enables the horizontal transmission of several viruses in natural and laboratory lepidopteran populations, including nucleopolyhedroviruses[14–16]. However, in the laboratory, the cannibalistic behavior in the rearing larval population may incur costs, forcing us to separate the insects and rear them individually, which would take more time and effort.

*Spodoptera exigua* (Hübner) is a globally distributed insect pest species, and its hosts have been recorded as 138 species belonging to 38 families, including corn, soybean, and other vegetables[17,18]. As a polyphagous insect pest, the high fecundity and insecticide resistance of *S. exigua* have been widely documented[19,20]. Convenient indoor maintenance of *S. exigua* colonies can provide a sufficient source of insects for study, regardless of insecticidal resistance development or biological control. For example, in our laboratory, a large number of noctuid larvae are required to maintain stocks of different insect viruses. As described above, during the daily rearing of noctuid larvae, different artificial diets were prepared and used[21,22], and different larval species showed different cannibalism frequencies. The larvae of *Helicoverpa armigera* (Hübner), *Helicoverpa assulta* (Guenee), and *Spodoptera frugiperda* were more likely to exhibit cannibalic behavior, especially in the 3rd instar and above, whereas the larvae of *Spodoptera litura* (Fab) and *Mythimna separata* (Walker) were seldom observed to exhibit cannibalic behavior, even when there was not enough diet in their rearing boxes. The *S. exigua* larvae showed varied cannibalism frequency when different diets were used. Before we found that feeding different diets led to a range of cannibalism frequencies in *S. exigua* larvae, the cannibalism of *S. exigua* larvae brought great trouble to our colonies, which led to the extinction of *S. exigua* colonies.

Different diets may have different nutrients, and due to their different origins, these might lead to different digestion and absorption by larvae after feeding and might show differences in insect growth and development, physiological and biochemical activities, feeding, mating or other behaviors, pupal weight, and female fecundity[23–26] as many studies have documented. As described above, diet quality can influence larval cannibalism. However, the precise effect of diet quality on larval cannibalism has not been well-studied. Larvae fed different diets may be associated with distinct microbial communities in the gut, and gut microbes are correlated with host nutrient metabolism, essential amino acid supplementation, and immunity[27–30]. Do diets fed to larvae affect larval cannibalism via the gut microbes? The present study was conducted to test this hypothesis. Our results provide a multi-dimensional perspective for understanding larval cannibalism.

## Results

**Developmental tests of *S. exigua* larvae fed with different diets.** The components of the artificial diet are listed in Table 1. The developmental periods for each immature stage and adult longevity when fed the six artificial diets are shown in Table 2. *S. exigua* could not complete its life cycle when fed Diet 6. Only one larva entered the 3rd instar stage when fed with Diet 6 but finally died. The immature (from the 1st to the 4th instars) and prepupal stages varied in the larvae fed with different artificial diets. There were no significant differences in pupal stage and adult longevity (in females and males) between larvae fed with different diets ($P > 0.05$).

The curves of the age-stage survival rate ($S_{xj}$) show the probability that a newborn will survive to age $x$ and develop to stage $j$ (Fig. 1A). The overlap of the stage-specific winner curves results from variable developmental rates among individuals. When larvae were fed Diets 1 and 5, high mortality rates were observed at the prepupal and pupal stages. To compare the development of larvae fed different diets, daily weight gain and consumption were measured from the 3rd instar (Fig. 1B, C) larvae. Diet 3-fed larvae had a significantly higher average weight

**Table 1 Formula of artificial diet for maintaining insect larvae (1 L).**

|  | Component group A[a] | Component group B | Usage |
|---|---|---|---|
| Diet 1 | Cornmeal (75 g), soybean powder (75 g), yeast (10 g), distilled water (700 ml), agar (20 g) | Sorbic acid (1 g), nipagin (2 g), vitamin C (2 g), cholesterol (0.1 g) | Test of *Spodoptera exigua* larvae in this study. |
| Diet 2 | Cornmeal (75 g), soybean powder (75 g), yeast (10 g), distilled water (700 ml), agar (20 g) | ketchup (200 g), Sorbic acid (1 g), nipagin (2 g), vitamin C (2 g), cholesterol (0.1 g) | Test of *Spodoptera exigua* larvae in this study. |
| Diet 3 | Rabbit grain (160 g), wheat germ slice (150 g), yeast (10 g), distilled water (700 ml), agar (30 g) | Sorbic acid (1 g), nipagin (3.8 g), vitamin C (8 g) | Laboratory maintaining larvae of *S. exigua*, *S. frugiperda*, *H. armigera*. |
| Diet 4 | Cornmeal (100 g), soybean powder (150 g), wheat bran (50 g), yeast (10 g), distilled water (700 ml), agar (25 g) | Sucrose (10 g), sorbic acid (5 g), vitamin C (5 g), cholesterol (1 g) | Test of *Spodoptera exigua* larvae in this study. |
| Diet 5 | Soybean powder (60 g), wheat bran (180 g), yeast (10 g), distilled water (700 ml), agar (25 g) | Sucrose (12 g), sorbic acid (4.8 g), nipagin (4.8 g), vitamin C (4.8 g), cholesterol (3 g) | Test of *Spodoptera exigua* larvae in this study. |
| Diet 6 | Wheat bran (180 g), yeast powder (10 g), distilled water (1000 ml), agar (16.8 g) | Sorbic acid (1 g), nipagin (3.8 g), vitamin C (8 g) | Test of *Spodoptera exigua* larvae in this study. |

[a]Cornmeal was purchased from XINXIANG HANGYU CORN PROCESSING Co., LTD.CHN; wheat bran was purchased from New hope LIUHE Co., LTD.CHN; Ketchup was purchased from Lee Kum Kee (XINHUI) Food Co., LTD.CHN; nipagin, vitamin, cholesterol and sorbic acid was purchased from Shanghai Macklin Biochemical Co., LTD.CHN; sucrose was purchased from JIANGSU LONGCEHNG FINE CHEMICAL Co., LTD.CHN; soybean powder was purchased from EASTOCEAN OLS GRAINS INOUSTRUES (ZHANGJIAGANG) Co., LTD.CHN; wheat germ slices was purchased from XINHUA HENGTAIYUAN FOOD Co., LTD.CHN; rabbit food was purchased from BEIJING KEAO XIELI FEED Co., LTD.CHN; yeast was purchased from Angel Yeast Co., LTD.CHN.

**Table 2 Developmental time, longevity, and mean fecundity of *Spodoptera exigua* on different artificial diets.**

| | Diet 1 | | Diet 2 | | Diet 3 | | Diet 4 | | Diet 5 | | Diet WY | |
|---|---|---|---|---|---|---|---|---|---|---|---|---|
| | n | Developmental time (d) (mean ± SE) | n | Developmental time (d) (mean ± SE) | n | Developmental time (d) (mean ± SE) | n | Developmental time (d) (mean ± SE) | n | Developmental time (d) (mean ± SE) | n | Developmental time (d) (mean ± SE) |
| 1st instar | 72 | 2.37 ± 0.12 c | 59 | 4.22 ± 0.10 ab | 66 | | 72 | 3.89 ± 0.07 ab | 72 | 3.79 ± 0.06 ab | 50 | 3.14 ± 0.19 d |
| 2nd instar | 60 | 3.68 ± 0.13 cd | 72 | 2.65 ± 0.21 b | 72 | | 48 | 4.38 ± 1.24 a | 48 | 2.58 ± 0.18 b | 72 | 5.00 ± 0.31 a |
| 3rd instar | 64 | 3.55 ± 1.38 d | 59 | 3.28 ± 0.06 bc | 46 | | 59 | 4.86 ± 0.28 b | 59 | | | 2.72 ± 0.11 b |
| 4th instar | 72 | 1.89 ± 0.07 c | | 3.53 ± 0.17 a | 72 | 2.70 ± 0.21 b | 47 | 2.42 ± 0.09 b | 44 | 2.94 ± 0.09 a | 1 | 4.00 ± 0.00 a |
| 5th instar | 71 | 3.29 ± 0.12 ab | | 2.68 ± 0.10 ab | 72 | | 44 | 2.25 ± 0.10 d | | | | 2.77 ± 0.13 bc |
| 6th instar | 23 | 2.63 ± 0.13 cd | | | | | | | | | | |
| Prepupa | 71 | 2.25 ± 0.12 ab / 2.43 ± 0.20 b | 59 | 9.16 ± 0.26 a | 38 | | | 2.94 ± 0.09 a | 44 | | | 2.24 ± 0.09 b |
| Pupa | 36 | 8.16 ± 0.14 a | 8 | 8.40 ± 0.24 a | 26 | | 15 | 9.57 ± 0.30 a | 39 | 9.35 ± 0.11 a | 5 | 9.35 ± 0.11 a |
| **Adult longevity (d) (mean ± SE)** | | | | | | | | | | | | |
| Female | 21 | 9.16 ± 0.34 a | 23 | 9.67 ± 0.67 a | 13 | 9.27 ± 0.14 a | 15 | 9.42 ± 0.16 a | 14 | 10.60 ± 0.24 a | | 10.00 ± 0.37 a |
| Male | 15 | 9.00 ± 0.38 a | 15 | 9.60 ± 0.33 a | | | 15 | 9.60 ± 0.24 a | 3 | 9.67 ± 0.33 a | | – |

Note: values followed by different lowercase letters within a row are significantly different using one-way ANOVA and by LSD analysis (P < 0.05).

and average daily consumption than the other diets at three days post-exposure (Fig.1B, C)

**Comparison of cannibalism of *S. exigua* larvae fed with different diets**. Because *S. exigua* larvae seldom survived into the 3rd instar with the maintenance of Diet 6, all of the following experiments were performed with larvae fed with the other five artificial diets (Diet 1, Diet 2, Diet 3, Diet 4, and Diet 5). The rates of cannibalism among *S. exigua* 3rd instar larvae fed with the five artificial diets were compared (Fig. 2). Five larval population densities (2, 5, 10, 15, and 20) were used to assess cannibalism. There was a significant difference in the ratio of cannibalism between larvae fed with different diets from 2 to 10 d post detection ($P < 0.05$, Supplementary Data 2). A heat map of the average cannibalism rates of larvae fed with different diets at different population densities was constructed (Fig. 2B). Diet 3 and Diet 5-fed larvae had a "blue square" in most cases, indicating that larvae fed with these two diets had low cannibalism rates. Figure 2A shows images of larvae fed different diets at a population density of 10 larvae at four days post-detection. The individual size of Diet 3-fed larvae was more uniform than that of larvae fed with other diets, and the bitten larvae (indicated by red arrows) or the dark stools pulled out after cannibalism (indicated by yellow arrows) could be easily distinguished. The cannibalism ratios of larvae fed with different diets under different larval population densities at ten days post-detection (all the tested larvae were dead or had pupated at this time) are provided in Fig. 2C. Diet 3-fed larvae had the lowest cannibalism ratio among the five tested larval groups, which was significantly lower than those of the Diet 1, Diet 2, or Diet 4-fed larvae ($P < 0.05$).

**Analysis of bacterial load of different diets maintained *S. exigua* larval guts**. The midguts of larvae fed five different diets (Diets 1, 2, 3, 4, and 5) were dissected (Fig. 3A) and used to determine the bacterial population and larval sex by real-time quantitative polymerase chain reaction (qPCR) [Fig. 3B, C]. Standard curves of the three primer pairs used for qPCR are shown in Supplementary Fig. S1. Among the larvae fed with different diets, Diet 3-fed larvae had the highest average midgut bacterial load; however, it was not significantly higher than that of Diets 1, 4, and 5, whereas it was significantly higher (approximately 100 fold) than that of Diet 2 fed larvae (Fig. 3B). To determine whether the bacterial load was correlated with larval sex, larval sex was determined using relative kettin-to-ATPase copies (Fig. 3C). The bacterial loads of the female and male larvae fed with the same diet were compared. There were no significant differences between female and male larvae in the bacteria in the midgut of larvae fed with the five diets ($P > 0.05$ [Fig. 3C]), indicating that feeding with Diet 2 reduced the larval midgut bacterial load and was not correlated with larval sex.

**Larval gut bacterial analysis based on 16S rRNA sequence**. To investigate the microbial community and structure in the midgut of 3rd instar *S. exigua* larvae fed with the five diets, sequences of the 16S rRNA gene V3–V4 variable regions were analyzed (Fig. 4). A total of 2,113,322 raw paired-end reads with an average length of 300 bp were obtained. After initial quality control, 2,080,588 high-quality sequences were obtained from 15 samples (three biological repeats for each dietary treatment, Supplementary Data 3). Based on 99% species similarity (Supplementary Data 4), 170, 307, 296, 256, and 394 operational taxonomic units (OTUs) were obtained from the midgut of larvae fed with Diets 1, Diet 2, Diet 3, Diet 4, and Diet 5, respectively (Supplementary Fig. S2, Supplementary Data 5, 6). There were 111 OTUs in all the samples, defined as common OTUs.

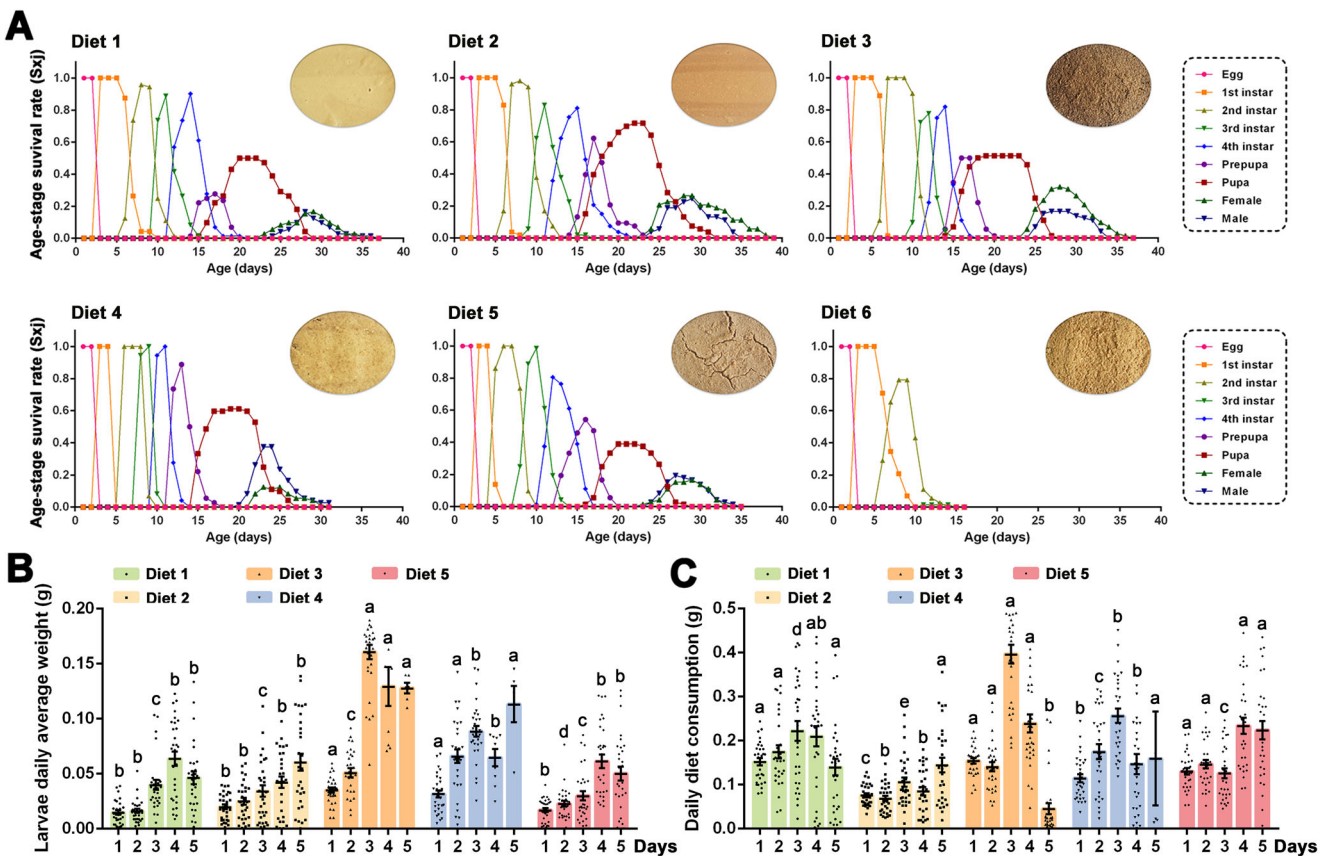

**Fig. 1 Effects of different diets on *Spodoptera exigua* growth and development. A** Age-stage specific survival rates of *S. exigua* on six artificial diets. The prepared diet images are on the upper right corner of each chart. **B** Daily average weight (mean ± SEM) of *S. exigua* larvae fed with Diet 1, Diet 2, Diet 3, Diet 4, or Diet 5 from the 3rd instar to prepupal stages. **C** Daily average diet consumption (mean ± SEM) of *S. exigua* larvae fed with Diet 1, Diet 2, Diet 3, Diet 4, or Diet 5 from the 3rd instar to prepupal stages. Different lowercase letters indicate statistical differences between means of larvae fed with different diets daily based on a one-way ANOVA followed by LSD comparisons (α = 0.05). The source data behind Fig. 1 were provided in Supplementary Data 1.

Twenty phyla were obtained from all samples, among which Firmicutes was the most predominant, accounting for >99% of the total phyla in the midguts of Diet 2, Diet 4, and Diet 5 fed larvae (Supplementary Data 7). Furthermore, in the midgut of larvae fed with Diet 1 and Diet 3, Proteobacteria accounted for ~20–40% of the total phyla. In contrast, Firmicutes and Proteobacteria combined accounted for >99% (Fig. 4A). At the class level, Bacilli was the most predominant class in the five dietary treatments. In contrast, Alphaproteobacteria and Gammaproteobacteria accounted for a considerable proportion of Diet 1 and Diet 3-fed larvae, respectively (Supplementary Data 8). At the order level, Lactobacillales were the most predominant among the total classes in all five diet-treated samples. In contrast, Rhodospirillales and Enterobacterales had considerable proportions in the midgut of Diet 1 and Diet 3-fed larvae (Supplementary Data 9). The Enterobacterales were the unique family in the midgut of Diet 3-fed larvae, accounting for ≤0.1% in the midgut of larvae fed with the other diets. At the family level, Enterococcaceae was the most predominant family in the midgut of larvae fed the five diets. In contrast, Acetobacteraceae and Enterobacteriaceae also had considerable proportions in the midgut of Diet 1 and Diet 3-fed larvae (Supplementary Data 10). At the genus level, *Enterococcus* was the most predominant among the five dietary treatments, whereas *Acetobacter* accounted for a considerable portion of Diet 1-fed larvae (Supplementary Data 11).

Linear discriminant analysis Effect Size (LEfSe) diagrams of Diet 1, Diet 2, Diet 4, and Diet 5 versus Diet 3 fed samples are shown in Fig. 4B, and the detailed information of the OTU taxonomic assignments of each comparison were provided in Supplementary Data 13–16. The LEfSe diagram of Diets 1 versus 3 revealed that Acetobacteraceae and Enterococcaceae were the most significantly enriched biomarker families. Several common biomarker families were identified in the other three LEfSe diagrams, including Rhodospirillales (red arrows in the phylum Proteobacteria), Enterococcaceae, and Leuconostocaceae (blue arrows in the phylum Firmicutes). The biomarker families Enterococcaceae and Leuconostocaceae were enriched in Diet 2 vs. Diet 3, Diet 4 vs. Diet 3, and Diet 5 vs. Diet 3, whereas Enterococcaceae were enriched in Diet 2, Diet 4, and Diet 5 treated samples, and Leuconostocaceae were enriched in Diet 3.

Furthermore, functional difference analyses of bacterial clusters of orthologous genes (COG) were performed, and the results are shown in Fig. 4C, and the detailed information of OTUs in wilcox test of each comparison were provided in Supplementary Data 17–20. It has four primary functions (metabolism, cellular, information, and poor). Among these, the metabolism group had the most abundant bacteria in the four comparisons (Diet 1 vs. Diet 3, Diet 2 vs. Diet 3, Diet 4 vs. Diet 3, and Diet 5 vs. Diet 5). Compared to the Diet 3-fed samples, Diet 1-fed samples had down-regulated relative abundance in "metabolism" and "cellular" functions and up-regulated relative abundance in

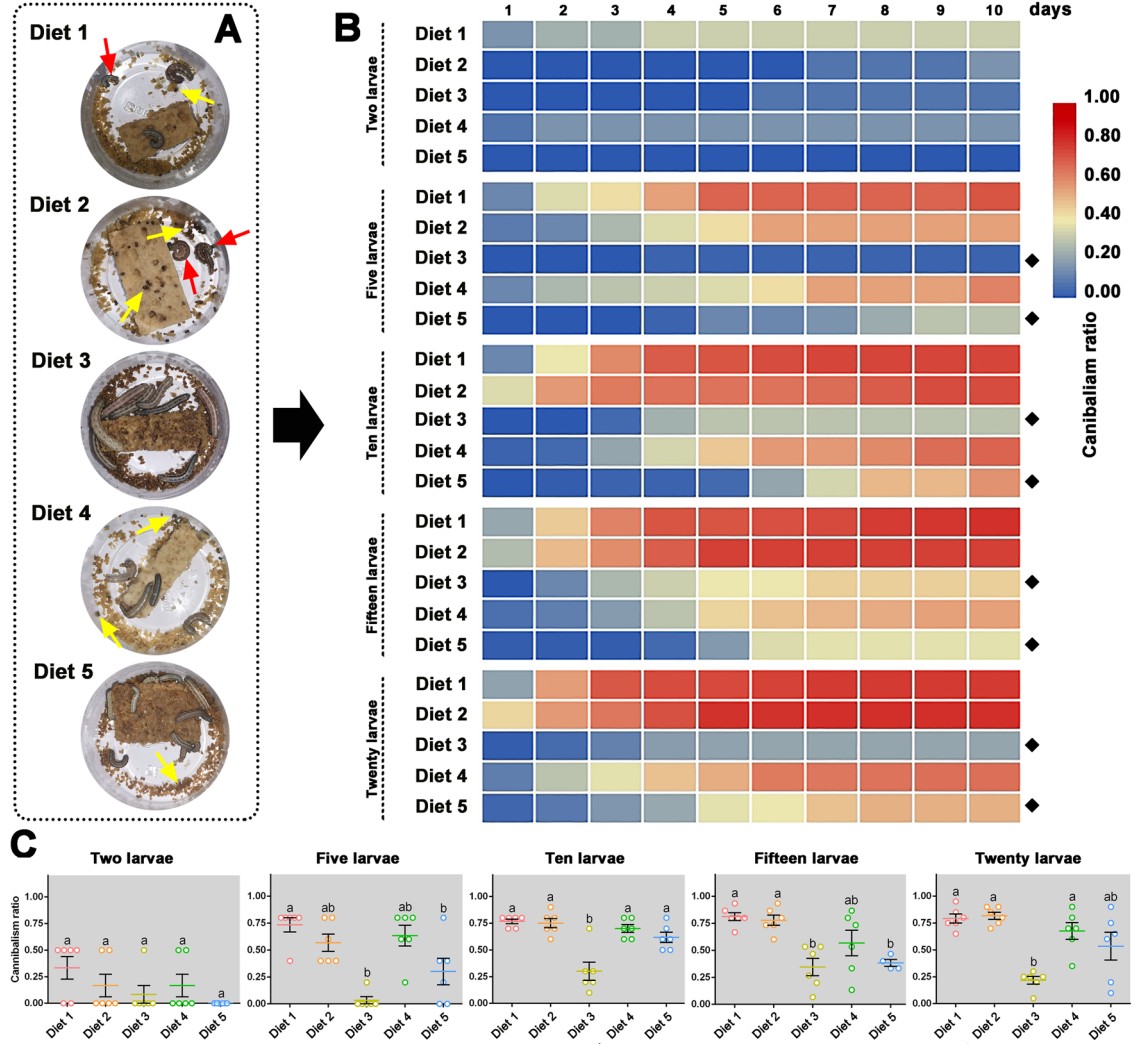

**Fig. 2 Cannibalism detection of 3rd instar *Spodoptera exigua* larvae fed with different artificial diets. A** Images of *S. exigua* larvae fed with different artificial diets in the bioassay chambers at four days post cannibalism detection. The red arrows indicate the cannibalized larvae and the yellow arrows indicate the dark stools pulled by the intraspecific predators. **B** The heat map of daily cannibalism ratios of *S. exigua* larvae fed with different diets under different population densities. The black diamond makers at the right panel indicate delayed or larval cannibalism inhibition. **C** The larval cannibalism ratio (mean ± SEM) of 3rd instar *S. exigua* larvae reared with the five diets under five larval densities at ten days post-detection. Different lowercase letters indicate statistical differences between means of the larvae fed with different diets based on a one-way ANOVA followed by LSD comparisons (α = 0.05). The source data behind Fig. 2 were provided in Supplementary Data 1.

"information" and "poorly" functions compared with those of samples from Diet 3-fed larvae; however, the regulations were opposite in the other three comparisons, indicating that the midgut of larvae fed with Diet 1 or Diet 3 contained more specific bacterial species compared with those of larvae fed with Diet 2, Diet 4, and Diet 5 diets.

**Bacterial isolates obtained from the midgut of Diet 3 cultured *S. exigua* larvae.** The results obtained from the above experiments suggested that Diet 3-fed *S. exigua* larvae showed inhibited cannibalic behavior, and gut bacterial analysis showed that Enterobacterales was a unique family in the midgut of Diet 3-fed larvae. Thus, the midgut of Diet 3-fed larvae was dissected and used to isolate Enterobacterales for further functional analyses. Fifteen bacterial isolates were obtained from the gut of Diet 3-fed *S. exigua* larvae, and their phenotypic and genotypic characteristics are shown in Supplementary Fig. S3. All the isolates were taxonomically distributed across the phyla Firmicutes and Proteobacteria. These isolates belonged to the genera *Bacillus*,

*Lysinibacillus*, *Escherichia*, *Enterococcus*, *and Mammaliicoccus* and represented 15 species. Macroscopic and microscopic characteristics of the isolates were determined to corroborate the genotypic identification (Supplementary Data 21). The biochemical characteristics of the bacterial isolates are presented in Supplementary Data 21. Based on phenotypic and genotypic analyses, the bacterial flora of *S. exigua* contained *Escherichia coli* (SePC-2, -20, -26, -33, -36, -37, and SeXC-25, -27, and -35), *Lysinibacillus sp.* (SePC-3), *Escherichia sp.* (SePC-12), *Bacillus cereus* (SePC-32), *Enterococcus mundtii* (SePC-38), *Bacillus sp.* (SeXC-34), and *Mammaliicoccus sp.* (SeXC-36). Partial 16S rRNA gene sequences were deposited in the GenBank database under the accession numbers OP363825, OP363838, and OP363926. Additionally, phylogenetic analysis matched the phenotypic and genotypic identifications (Supplementary Fig. S4).

**Effects of bacterial loading on the *S. exigua* larval cannibalism.** Fifteen bacteria isolated from the midgut of the Diet 3-fed 3rd instar *S. exigua* larvae were loaded into other diets to test whether

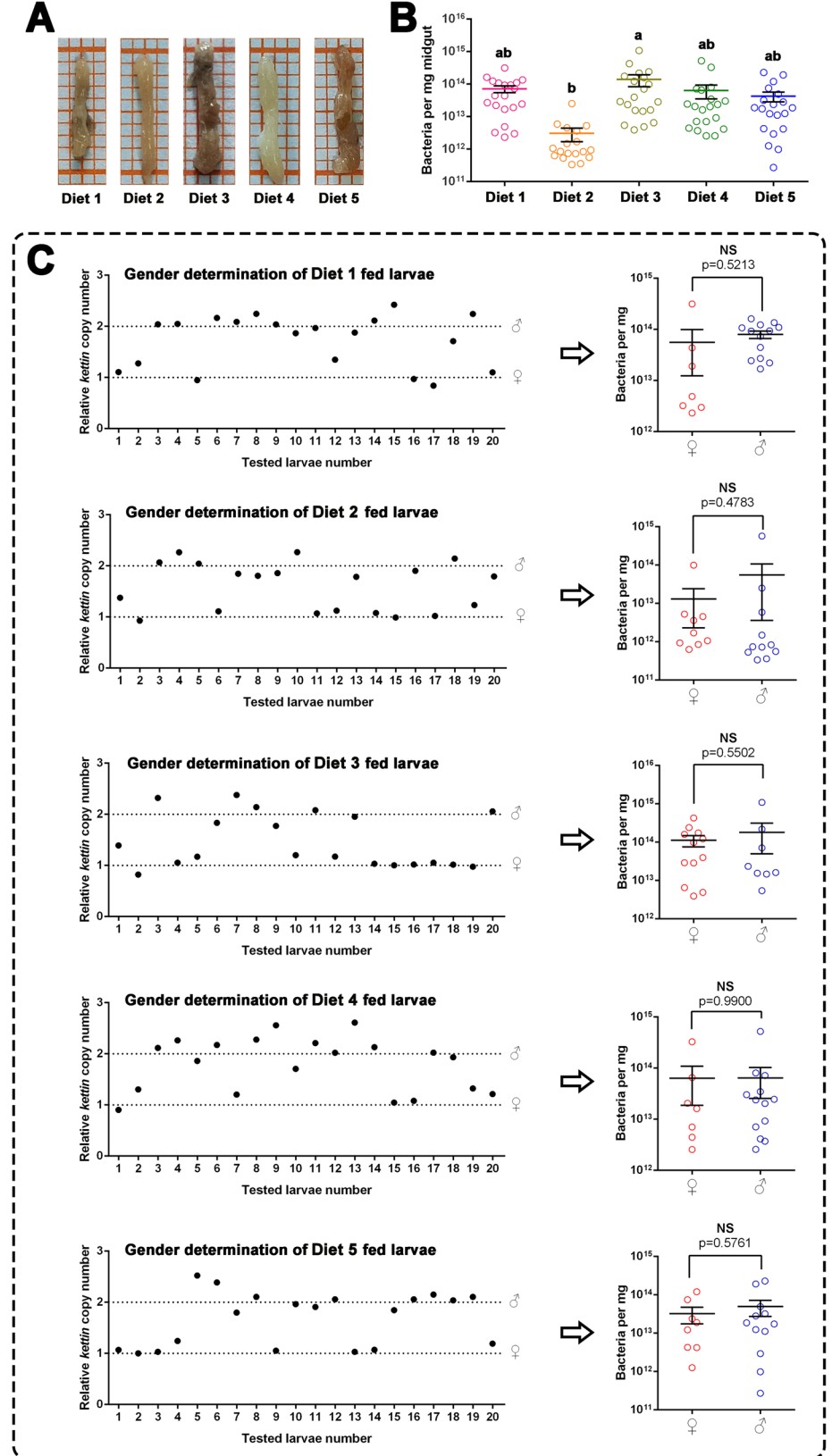

these bacteria had any effects on the cannibalism of larvae maintained with other diets. The COG functional difference analysis of the comparisons between Diet 3-fed larvae and the other four diet-fed larvae indicated that the gut bacterial function of Diet 1-fed larvae might differ from that of the other diet-fed larvae (Fig. 4C). Diets 1 and 2 had almost the same composition;

the only difference was that the latter contained ketchup, whereas the former did not. Thus, in the following experiments, we chose Diet 1- and Diet 2-fed larvae to detect the possible effects of the bacteria isolated from Diet 3-fed larvae (Fig. 5A). Compared with Diet 1 dipped in Luria Broth (LB) medium (CK), Diet 1 dipped in *SePC-2, -12, -26, -32, -36, or -37* delayed cannibalism, among

**Fig. 3 Influence of diets or sex on the midgut bacterial load of *Spodoptera exigua* larvae. A** The images of the dissected midgut of 3$^{rd}$ instar *S. exigua* larvae fed with Diet 1, Diet 2, Diet 3, Diet 4, or Diet 5. **B** Comparison of bacterial load (mean ± SEM) in the midgut of *S. exigua* larvae reared with five diets. Absolute quantification polymerase chain reaction (PCR) was used to determine the bacterial number in each midgut sample. Different lowercase letters indicate the differences between means from larvae fed with different diets based on one-way ANOVA followed by LSD comparisons (α = 0.05). **C** Sex determination of the *S. exigua* larvae (left panel) and bacterial load comparison between the female and male larvae fed with five diets (right panel). The Student's *t* test was used to compare the differences between the bacteria load in the midguts of female and male larvae. The source data behind Fig. 1 were provided in Supplementary Data 3.

which the addition of *SePC-12* and *-37* maintained the tested larvae at a relatively low cannibalism ratio from 1 to 4 days post detection (Fig. 5B). Compared with Diet 2 dipped in LB medium, Diet 2 dipped in *SePC-12* and *-26 and SeXC-25* cultures delayed cannibalism, among which the addition of *SePC-12* lowered the cannibalism ratio of the tested larvae from one to six days post detection (Fig. 5C). Detailed data on the daily average larval cannibalism ratios are provided in Supplementary Data 22 and 23.

Furthermore, bacterial colonization in the larval midguts after feeding was detected by determining the total bacterial load changes using qPCR (Fig. 5D). After loading a specific bacterial isolate, the total bacterial content increased. However, significant differences were only observed between the LB load and *SePC-12* loaded samples in Diet 1-fed larvae [*F*-value (*F*) = 2.623, degrees of freedom (DF) = 7, 124, and *P*-value (*P*) = 0.0147] and the LB load and *SeXC-35* loaded samples in Diet 2-fed larvae (*F* = 2.258, DF = 7, 124, and *P* = 0.0340) at 24 h post-bacterial loading. However, there were significant differences between the LB load samples and any other bacterial-loaded samples in Diet 1-fed larvae (*F* = 2.315, DF = 7, 124, and *P* = 0.0301) and Diet 2-fed larvae (*F* = 4.165, DF = 7, 124, and *P* = 0.0004), indicating that the bacteria colonized the larval midguts after loading. Furthermore, the change in the cannibalism ratio at the detection points shown in Fig. 5B, C may have resulted from specific bacterial loading.

**Battles between different diet-fed larvae and insect hormone detection.** The 3$^{rd}$ instar larvae fed Diet 1, Diet 2, and Diet 3 were starved separately and then transferred into a plastic cup without food to facilitate in-group (two larvae transferred from the same diet-fed population) and cross-group battles (two larvae transferred from different diet-fed populations [Fig. 6A]). After starvation for 24 h (before battles [control]), larvae from the Diet 2-fed group had significantly lower juvenile hormone (JH) concentrations than those of larvae fed Diets 1- or Diet 3-fed groups (*P* < 0.05), whereas there were no significant differences between the ecdysone concentrations of larvae from the Diet 1, Diet 2, or Diet 3-fed groups (Fig. 6B). After cannibalism, the winners were collected, and insect hormone concentrations were determined. After in-group battles, the winners of the Diet 2-fed larvae had significantly increased JH concentration (about 1735.0 ± 730.0 ng/L/g larvae), which was significantly higher than that of other samples (*P* < 0.05). In addition to the winners from Diet 1 vs. Diet 1 battles, other winners, regardless of in-group or cross-group battles, had relatively lower JH concentrations, and there were no significant differences between the winners (*P* > 0.05). The winners in Diet 1 vs. Diet 3 battles had increased ecdysone concentrations, which were significantly higher than those of the winners in Diet 1 vs. Diet 1, Diet 1 vs. Diet 2, Diet 2 vs. Diet 2, and Diet 2 vs. Diet 3 battles (*P* < 0.05).

In the Diet 1 vs. Diet 2 and Diet 2 vs. Diet 3 battles, the winners were mainly Diet 2-fed larvae (Fig. 6C), whereas the winners in the Diet 1 vs. Diet 3 battle were either Diet 1 (seven larvae) or Diet 3-fed larvae (five larvae). Furthermore, the starting time of cannibalism in the cross-group battles ranged from 24.9 to 28.5 h,

which is longer than that of the in-group battles (ranging from 17.6 to 23.3 h). There were significant differences between the three cross-groups and Diet 2 vs. Diet 2 and Diet 3 vs. Diet 3 (*P* < 0.05[Fig. 6D]).

**Bacterial loading effects on *S. exigua* larval hormone.** Among the 15 bacterial isolates obtained, *SePC-37* had the most significant impact on Diet 1-fed larval cannibalism (Fig. 5B), whereas *SePC-12* had the most significant effect on Diet 2-fed larval cannibalism (Fig. 5C). Thus, *SePC-12* and *-37* were smeared on diet dots (Diet 1 or Diet 2) to feed *S. exigua* larvae. The larvae were collected 24 h and 48 h after the bacteria were added to determine the insect hormone concentration (Fig. 7A). After bacterial loading, there were no significant differences in insect ecdysone concentrations between the *SePC-12* and *-37* smeared diets (*P* > 0.05) (Fig. 7B). There was no significant difference between the JH concentration of the *SePC-37* loaded larvae and that of the starved larvae at 24 or 48 h post-loading in Diet 1- and Diet 2-fed larvae (*P* > 0.05). However, there was a significant increase in JH concentration in *SePC-12* loaded larvae 24 h post-loading compared to that in starved larvae in the diet 2-fed larval group (*F* = 3.246, DF = 4, 23, *P* = 0.0300). In contrast, there was no significant increase in the JH concentration in the Diet 1-fed larvae (*F* = 0.9445, DF = 4, 15, *P* = 0.4654).

**Comparison of cannibalism of three noctuid larvae after the loading of *SePC-12* and *SePC-37*.** To investigate whether the *SePC-12* and *SePC-37* had any effects on larval cannibalism (delay or inhibit host larval cannibalism) on other noctuid larvae, the larvae of fall armyworm (*Spodoptera frugiperda*), cotton ball worm (*Helicoverpa armigera*), and tobacco budworm (*Helicoverpa assulta*) were used to perform the detection (Fig. 7C). In contrast to *S. exigua*, these three pest species showed higher frequencies of cannibalism during laboratory rearing. In the present study, they were reared individually from the 3rd instar in the laboratory. In *S. frugiperda* and *H. armigera*, cannibalism began on the 2$^{nd}$ day of the test. Diet 2-fed larvae (regardless of whether bacteria were loaded) had more red squares than Diet 1-fed larvae of *S. frugiperda* and *H. armigera* (Fig. 7D), indicating that feeding Diet 2 resulted in a higher cannibalism ratio than feeding Diet 1 in the two species. The cannibalism ratio of the tested *H. assulta* larvae was lower than that of *S. frugiperda* and *H. armigera*, which began on the 3$^{rd}$ day of detection. Compared to larvae fed with LB medium-smeared diets, there were delayed larval cannibalism in *SePC-12* or *-37* smeared Diet 1-fed *S. frugiperda* larvae and *SePC-12* or *-37* smeared Diet 2-fed *H. assulta* larvae but the loading of *SePC-12* resulted in a more obviously delayed cannibalism behavior (marked with black arrows) compared to that of *SePC-37*.

**Determination of aggression and JH concentration of *SePC-12* mono-associated *S. exigua* larvae.** To confirm whether *SePC-12* could grow under alkaline conditions, *SePC-12* and *E. coli* TG1 were inoculated into LB medium with pH values ranging 7.5–11.0 to compare their growth (Fig. 8A). Both *SePC-12* and *E. coli* TG1

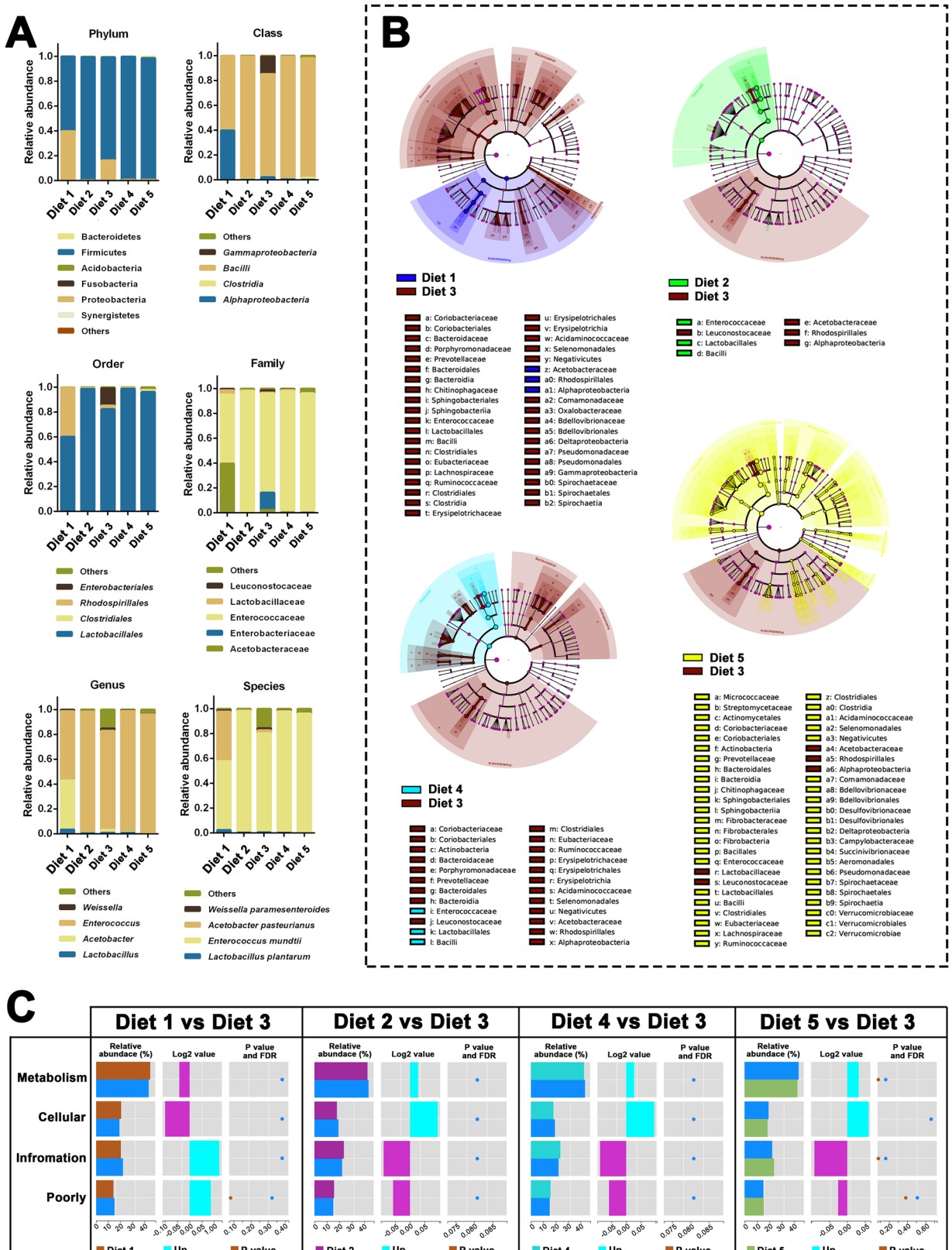

grew in all the tested LB media, and the optical density of *SePC-12* was significantly higher than that of *E. coli* TG1 from 16 to 72 h in the pH 7.5 to pH 9.5 LB medium, and significant differences were found between 12–72 h, 8–72 h, and 4–72 h at pH 10.0, 10.5, and pH 11.0 LB medium, respectively.

Germ-free (GF) *S. exigua* larvae were maintained to test the function of *SePC-12* (Fig. 8B). After 12 h, the GF larvae exposed to *SePC-12* contained Diet C, and the total bacterial content increased approximately 100 times, which was significantly higher than that of the GF larval gut ($t = 2.197$, df = 18, $P = 0.0413$,

**Fig. 4 Bioinformatic analysis of midgut microbiota of *Spodoptera exigua* larvae fed with different diets. A** Phylum-, class-, order-, family-, genus-, and species-level abundance of the dominant bacteria in the midguts of *S. exigua* larvae fed with different diets. The detailed information of relative abundance of OTUs in different samples at Phylum-, class-, order-, family-, genus-, and species-level were provided in Supplementary Data 7–12. **B** Linear discriminant analysis Effect Size (LEfSe) clustering tree of the gut samples of larvae fed with Diet 3 compared with those fed with other diets. Different colors represent different groups. Nodes with different colors represent microbiota that play an essential role in the groups represented by the same color. A color circle represents a biomarker. The legend below the tree is the name of the biomarker. The yellow nodes represent microbial groups that did not play a role in the different groups. Each circle is the level of phyla, class, order, family, and genus from the inside to the outside. The red arrows indicate the common biomarker family (Rhodospirillales) in the four LEfSe clustering trees. Blue arrows indicate the opposite clustered biomakers in the phylum Firmicutes (Enterococcaceae and Leuconostocaceae). **C** Wilcox Test of pathway differences of the gut samples fed with Diet 3 compared to those fed with the other diets. In each comparison, the left panels are the relative abundance of pathways in each group; the middle panels are the log2 values of the mean ratios of relative abundances of the same pathway in the two groups; the right panels show the *P*-value and false discovery rate (FDR) values obtained from Wilcox Test. Significant differences between the two groups were determined when the *P*-value and FDR value were less than 0.05.

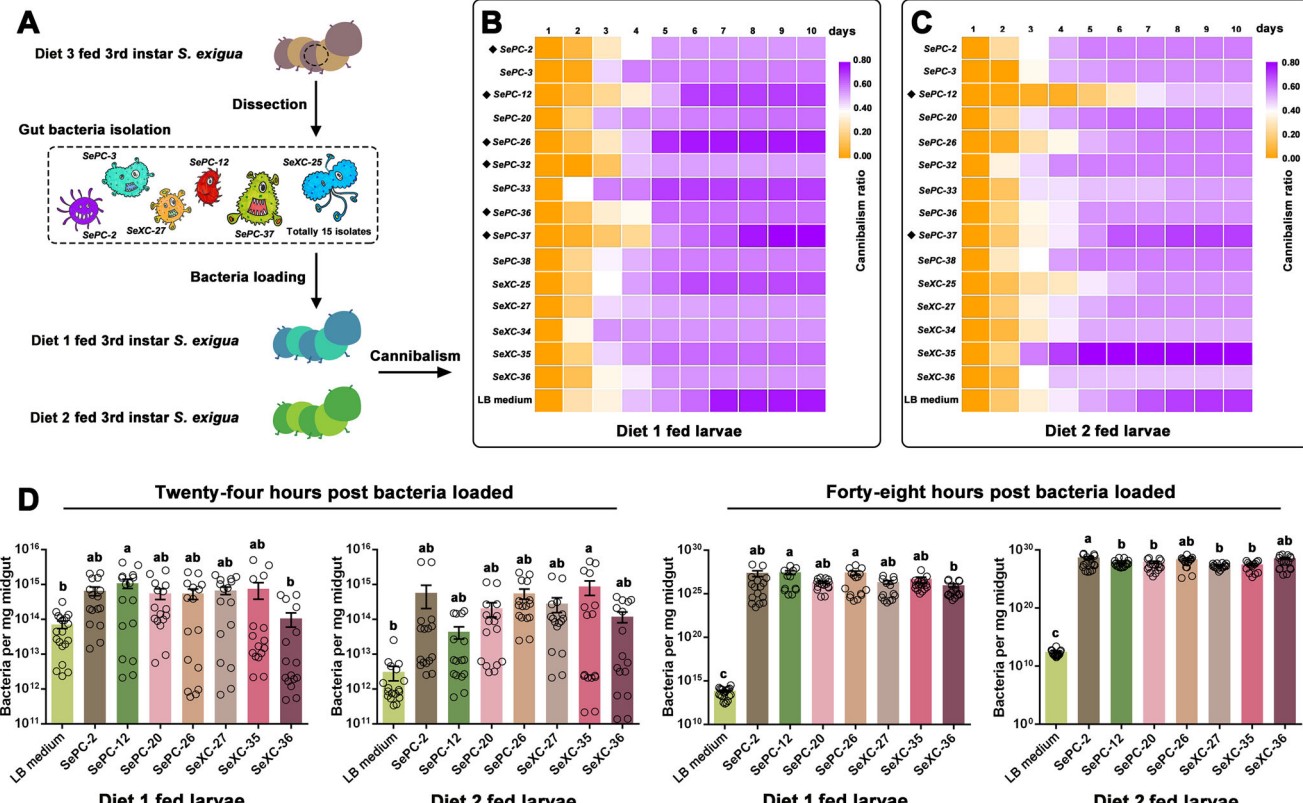

**Fig. 5 Cannibalism detection and bacterial load determination of *Spodoptera exigua* larvae fed with Diet 1 or Diet 2 after adding the gut bacteria isolates. A** The workflow of the bacterial isolation, bacterial addition, and cannibalism detection. **B** The heat map of the daily cannibalism ratio of 15 bacterial isolates added to Diet 1 and fed *S. exigua* larvae. **C** The heat map of daily cannibalism ratios of 15 bacterial isolates added to Diet 2 and fed *S. exigua* larvae. LB medium-smeared diet slices fed to *S. exigua* larvae were used as control (CK) during the detection. The black diamond makers in front of the bacterial names indicate delayed or inhibited larval cannibalism. **D** Determination of bacterial load after the specific bacterial isolates were added. Different lowercase letters indicate statistical differences of means between different bacterial isolates in the midgut of *S. exugia* larvae based on one-way ANOVA followed by LSD comparisons (α = 0.05). The source data behind Fig. 5 were provided in Supplementary Data 1.

Fig. 8C). The juvenile hormone (JH) concentration of GF *S. exigua* larvae was approximately 661.2 ± 163.6 ng/L/g larva, and after starvation for 24 h, the JH concentration of GF-SV *S. exigua* larvae was slightly decreased. However, the JH concentration significantly increased after the loading of *SePC-12* (F = 8.198, df = 5, 36, P < 0.0001), which was more than twice that of the GF *S. exigua* larvae (Fig. 8E). The cannibalism start time of *SePC-12* mono-associated *S. exigua* larvae (IGSEPC12) ranged from 24 to 40 h, which was slightly later than that of starved GF *S. exigua* larvae (Fig. 8D), and the start time for starved GF *S. exigua* larvae and *SePC-12* mono-associated larvae (CGGF/SePC12) ranged from 12 to 17 h, which was significantly earlier than that of the other two in-group battles (F = 10.72, df = 5, 10, P = 0.0073). In

addition, five of the winners were the GF-SV larvae in six cross-group battles (CGGF/SePC12). The winners from the three performed group battles had similar JH contents, which were significantly lower than that of *SePC-12* mono-associated *S. exigua* larvae but not significantly different from that of GF or GF-SV larvae had (F = 8.198, df = 5, 36, P < 0.0001).

## Discussion

Cannibalism has been widely reported in many species of lepidopteran larvae[31–36]. This study demonstrates the cannibalism of laboratory populations of *S. exigua* larvae under different artificial dietary supplies. Sufficient food was provided in this study. *S. exigua* larvae fed Diet 2 had a significantly higher cannibalism

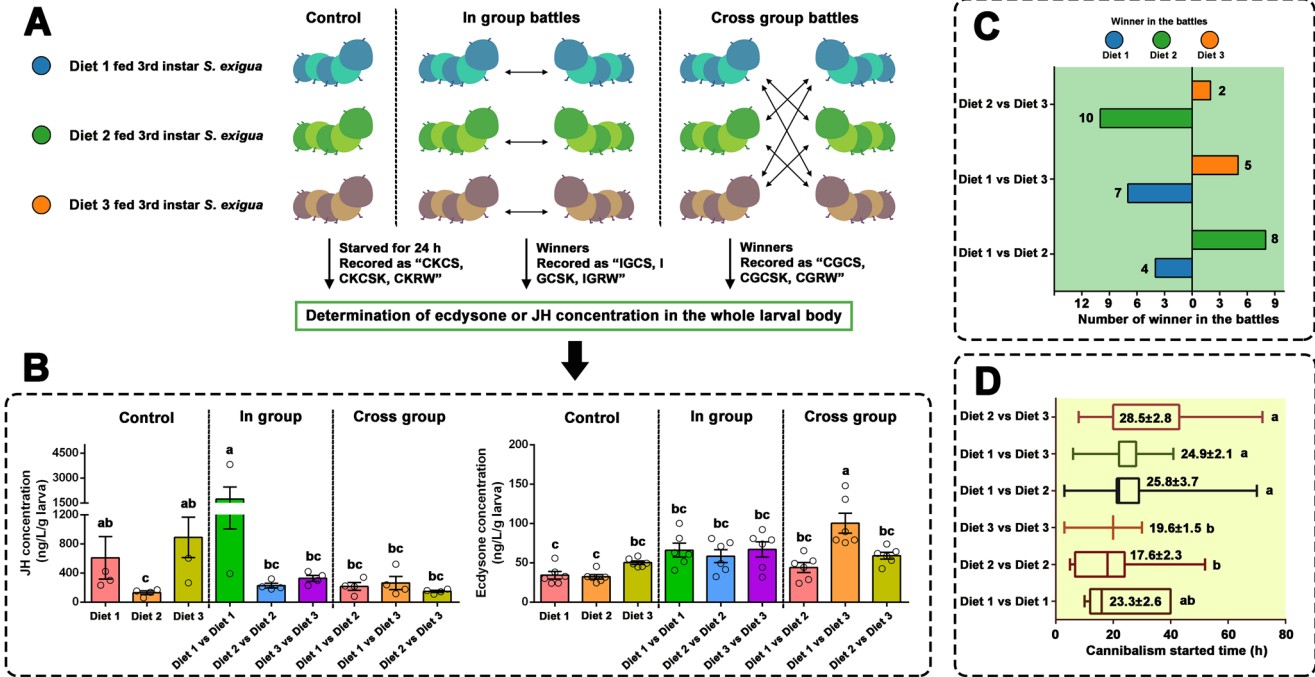

**Fig. 6 Larval in-group or cross-group battles and insect hormone determination. A** A diagram of larval treatment and illustration of collections of different samples. **B** Determination of larval juvenile hormone (JH) and ecdysone concentrations. The JH or ecdysone concentration (mean ± SEM) of the starved larvae (CK), and the winners in the in-group or cross-group battles were determined. **C** Number of winners in the cross-group battles. **D** The cannibalism starting time (mean ± SEM) in the in-group and cross-group battles. Different lowercase letters indicate statistical differences between means of different tested larvae based on one-way ANOVA followed by LSD comparisons (α = 0.05). The source data behind Fig. 6 were provided in Supplementary Data 1.

ratio than those fed with Diet 3- or Diet 5-fed larvae (Fig. 2). Diet 2-fed larvae had the lowest bacterial load in the midgut compared to larvae fed other feeds (Fig. 3). A previous study revealed that the bacterial load in the larval gut is associated with baculovirus infection, as the gut microbiota can benefit viral virulence, pathogenicity, and dispersion[37], and induce apparent immune priming, thus increasing host larval tolerance to *Bacillus thuringiensis*[38].

Taxonomic analysis of our study revealed that the *S. exigua* gut had the most common phyla of bacteria, which were also reported in other lepidopteran species[39–46]. Besides, our results suggesting that diet could influence the microbial gut composition of the *S. exigua* larvae, consistent with study reported by Gao et al.[44]. Gut microbiota are reported to be involved in various physiological functions, including digestion and protection against pathogenic infections[47], and our study suggests that gut bacteria are associated with the manipulation of host cannibalistic behavior. The correlation between gut bacteria and host cannibalistic behavior will likely be indirectly manipulated. Some gut bacteria modulate the host endocrine system and development[48,49]. Symbiotic bacteria in the hemolymph[50–52], which might be translocated from the gut[53–55], can also be influenced by diet[56]. These hemolymph-containing bacteria maintain homeostasis in their hosts and are recognized by host pattern recognition receptors (PRRs) via pathogen-associated molecular patterns[57]. The H. armigera C-type lectin (PRRs) *H. armigera*-mediated homeostasis of *E. mundtii* in the hemolymph is critical for normal larval growth and development[58]. Bacterial homeostasis in insect hemolymph is associated with humoral immunity in insects. The steroid hormone 20-hydroxyecdysone positively regulates innate immunity[59–61], whereas JH is an immune-suppressor[62] and may indirectly influence the association between symbiotic insect bacteria and host larval growth and

development, which is consistent with previous studies[48,49]. In the present study, the JH concentration in Diet 2-fed larvae was low (Fig. 7), and in cross-group battles, Diet 2-fed larvae predated on Diet 1- or 3-fed larvae to survive. The addition of *SePC-12* increased the JH concentration in Diet 2-fed larvae and delayed cannibalistic behavior (Fig. 7). It has been reported that larval cannibalistic behavior is correlated with JH concentration; *S. exigua* larvae fed with a juvenile hormone analog (JHA) had a decreased cannibalism ratio compared to that of non-JHA-fed larvae[63,64]. Baculovirus-infected larvae (usually considered to have a higher JH concentration than healthy larvae)[65–68] have a lower cannibalism ratio than healthy larvae. Baculovirus-infected larvae are more likely to be victims of cannibalism than healthy individuals, when mixed with healthy larvae[64].

The proposed effects of gut bacteria on host larval cannibalism are shown in Fig. 9. Different diets resulted in differences in larval growth, development, and aggression. A bacterium (*SePC-12*) isolated from the non-aggressive *S. exigua* larval group (Diet 3-fed larvae with a low cannibalism ratio) inhibited or delayed larval cannibalistic behavior after colonization by aggressive larval groups (especially Diet 2-fed larvae). Further investigation revealed that the JH concentration was lower in aggressive larvae, whereas JH concentration increased after they were fed a diet containing *SePC-12* (Fig. 9). The inhibitory function of *SePC-12* was also confirmed in Diet 1- and Diet 2-fed *S. frugiperda*, *H. armigera*, and *H. assulta* larvae, among which loading of *SePC-12* delayed larval cannibalism in Diet 1-fed *S. frugiperda* and Diet 2-fed *H. assulta* (Fig. 7C, D). No significant changes in cannibalism of *H. armigera* larvae were found after the loading of *SePC-12*. These results indicated that the inhibition of larval cannibalism caused by the loading of *SePC-12* was limited, and the factors limiting the function of *SePC-12* require further study.

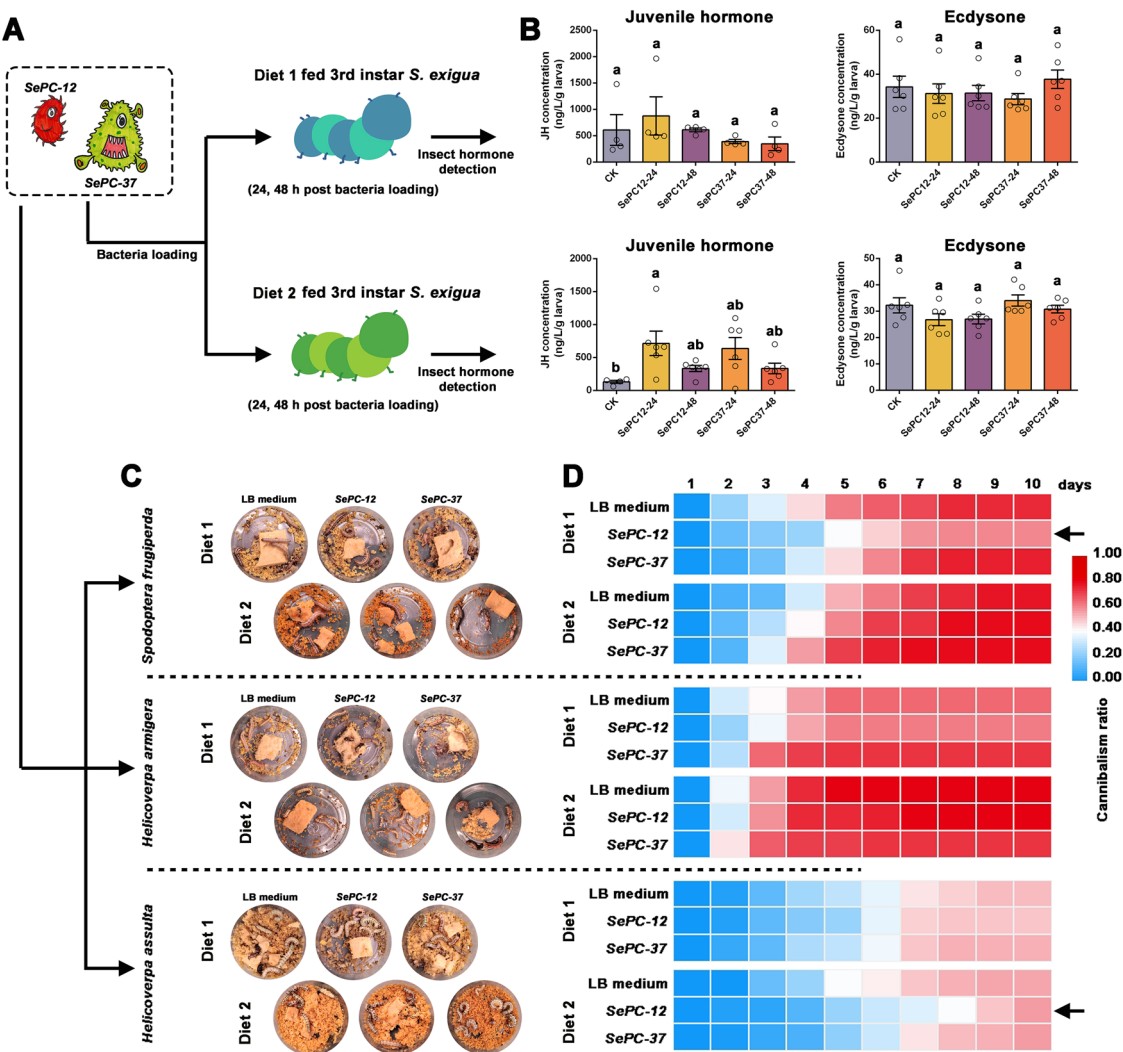

**Fig. 7 Functional confirmation of *SePC-12* and *SePC-37* in *Spodoptera exigua*, *Spodoptera frugiperda*, *Helicoverpa armigera*, and *Helicoverpa assulta* larvae. A** The workflow of the treatment of different samples. **B** Determination of juvenile hormone (JH) and ecdysone concentration of the 3rd instar *S. exigua* larvae after loading *SePC-12* or *SePC-37*. The starved larvae collected before the bacteria was added were used as CK. **C** Images of *S. frugiperda*, *H. armigera*, and *H. assulta* larvae fed with *SePC-12*, *SePC-37*, or LB medium (used as control) smeared diet slices. **D** The heat map of daily cannibalism ratios of different diets smeared with *SePC-12*, *SePC-37*, or LB medium (used as control) and fed to *S. frugiperda*, *H. armigera*, and *H. assulta* larvae. The black arrows indicate delayed or inhibited larval cannibalism in the tested *S. frugiperda* or *H. assulta* larvae loaded with bacteria. Different lowercase letters indicate statistical differences between the means of different tested larvae based on a one-way ANOVA followed by LSD comparisons ($\alpha = 0.05$). The ANOVA of the daily cannibalism ratio of *S. frugiperda*, *H. armigera*, *H. assulta* larvae fed Diet 1 or Diet 2 supplied with *SePC-12* and *SePC-37* were provided in Supplementary Data 24. The source data behind Fig. 7 were provided in Supplementary Data 1.

In conclusion, a possible correlation between midgut microbiota and larval cannibalism was observed in the present study. We assumed that *S. exigua* larvae employ bacterial symbionts and stabilize their hormone levels to regulate their cannibalistic behavior in the population. The results obtained in this study enrich the understanding of insect gut microbial function and also provide a method for reducing the cannibalic behavior in the mass rearing of noctuid larvae—proper addition of *SePC-12* in diets can allow us to increase the density of insect rearing and reduce the workload in daily insect rearing.

## Materials and methods

**Artificial diet preparation and insects rearing.** Six artificial diets (Diets 1, 2, 3, 4, 5, and 6) were prepared according to the formulas provided in Table 1. The ingredients of the artificial diets were divided into two groups: Group A provided the primary nutrients for larval growth, and Group B included preservatives and additives

to prevent spoiling of the diet. To prepare the artificial diets, the powders in group A were weighed and mixed gently in distilled water at 37 °C, followed by fermentation in an incubator (Sigma-Aldrich, St Louis, MO, USA) at 37 °C for 20 min. The fermented group A was then transferred into the autoclave at 121 °C for 20 min for sterilization. Distilled water was added to the sterilized group A to adjust the volume to 1 L. The weighed ingredients contained in group B were then added to the chilled group A (between 50–60 °C) and mixed thoroughly. The diets were transferred to a sterilized container and stored at 4 °C. The different diets were prepared separately. To ensure the effectiveness of the nutrients in the diet, the quality guarantee period of the prepared artificial diet did not exceed ten days. The contents of the major nutrients in each diet were calculated according to their contents in the commercial feed products (Supplementary Fig. S5).

For daily rearing and population reproduction, laboratory colonies of *S. exigua* were fed with Diet 3 at 27 ± 1 °C, 70 ± 5%

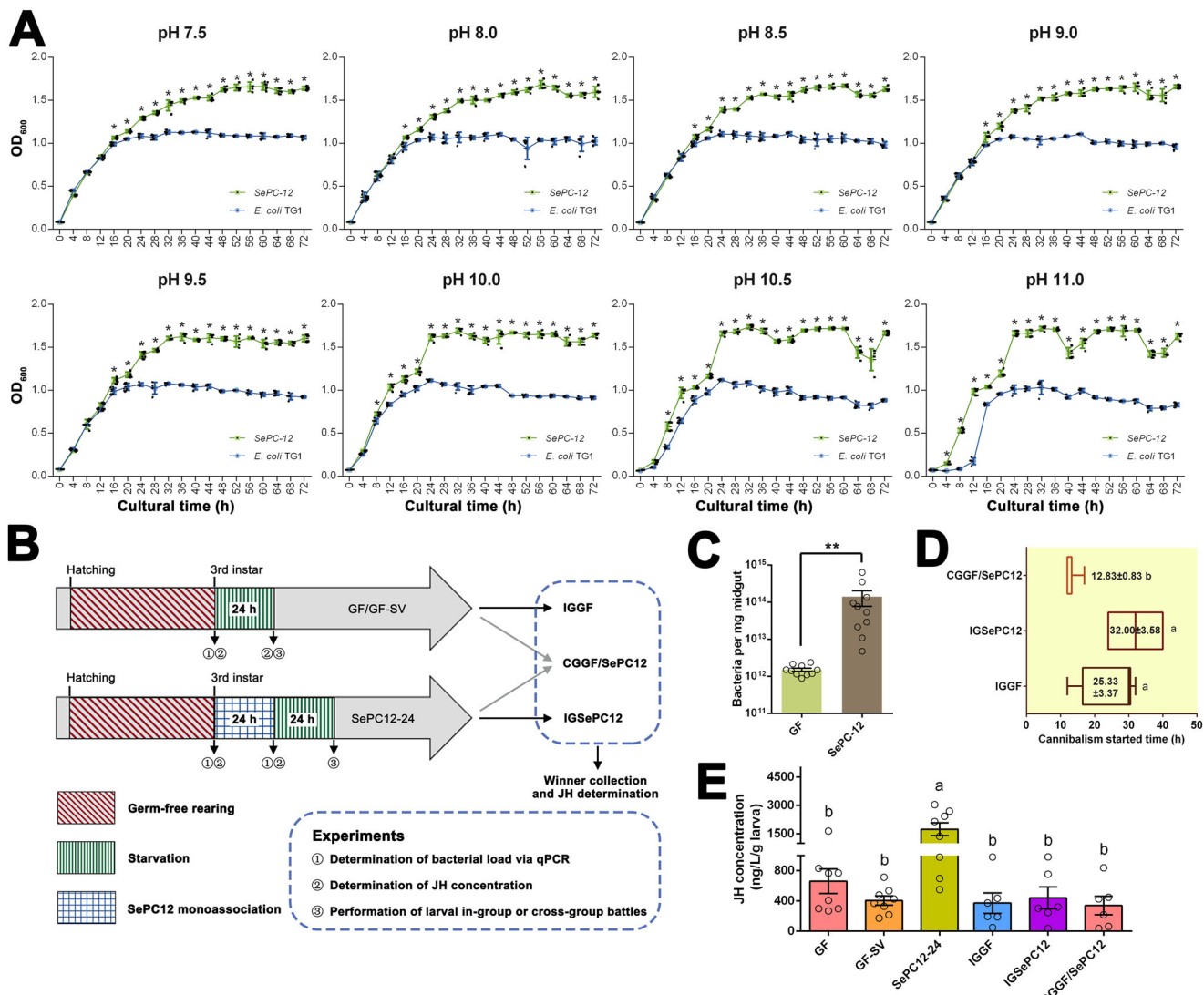

**Fig. 8 Determination of growth condition of *SePC-12* and effects of *SePC-12* on germ-free (GF) *S. exigua* larvae. A** Determination of growth condition of *SePC-12*. Asterisks indicate significant differences ($\alpha = 0.05$) between the optical density of *SePC-12* and *E. coli* TG1 at absorbance of 600 nm at each tested time based on Student's *t* test. **B** Experimental design showing *S. exigua* development, treatment, and experiments with GF and *SePC-12* mono-associated gnotobiotic *S. exigua* larvae. **C** Determination of the bacterial load in the midgut of GF and *SePC-12* mono-associated gnotobiotic *S. exigua* larvae. Asterisks indicate significant differences ($\alpha = 0.01$) between the bacterial load in the midgut of GF and *SePC-12* monoassociated gnotobiotic *S. exigua* larvae based on Student's *t* test. **D** The cannibalism starting time (mean ± SEM) of the in-group and cross-group battles of GF and *SePC-12* mono-associated gnotobiotic *S. exigua* larvae. **E** Determination of larval juvenile hormone GF and *SePC-12* mono-associated gnotobiotic *S. exigua* larvae and winners in in-group and cross-group battles. Different lowercase letters indicate statistical differences between means of different tested larvae based on one-way ANOVA followed by LSD comparisons ($\alpha = 0.05$). The source data behind Fig. 8 were provided in Supplementary Data 1.

relative humidity (RH), and a photoperiod of 16:8 h (L:D). Laboratory colonies of *S. frugiperda* and *H. armigera* were fed with Diet 3 under the same conditions as *S. exigua*. *H. assulta* larvae were collected from tobacco fields in Changsha City, China, and reared on a WS diet in the laboratory under the abovementioned conditions. Laboratory colonies of *S. exigua*, *S. frugiperda*, and *H. armigera* were maintained for more than 30 generations, whereas field-collected *H. assulta* colonies were maintained for more than ten generations.

**Analysis of *S. exigua* survival, growth, and development**. Newly hatched *S. exigua* larvae were transferred to fresh diets and reared on different diets (Diets 1, 2, 3, 4, 5, and 6). Twenty-four larvae were used for each replicate, and three replicates were performed for each diet. The dots were replaced every 48 h. The larvae were observed

daily, and dead larvae were recorded until death or pupation. The survival rates of larvae and larval instars were recorded daily. The mortality rate of each larval instar was calculated by dividing the number of dead larvae of the current instar by the number of surviving larvae of the previous instar. The life history data of all tested larvae were analyzed based on an age-stage two-sex life table[69] constructed using the TWO-SEX-MSChart program (National Chung Hsing University, Taichung, Taiwan)[70]. Differences in the developmental time of each larval instar, pupal period, and adult longevity were analyzed using one-way analysis of variance (ANOVA), and the least significant difference (LSD) was used to separate means in SPSS v22.0 (IBM Inc., Chicago, IL, USA).

**Detection of larval cannibalism**. To determine larval cannibalism, newly molted (within 12 h) 3[rd] instar *S. exigua* larvae from

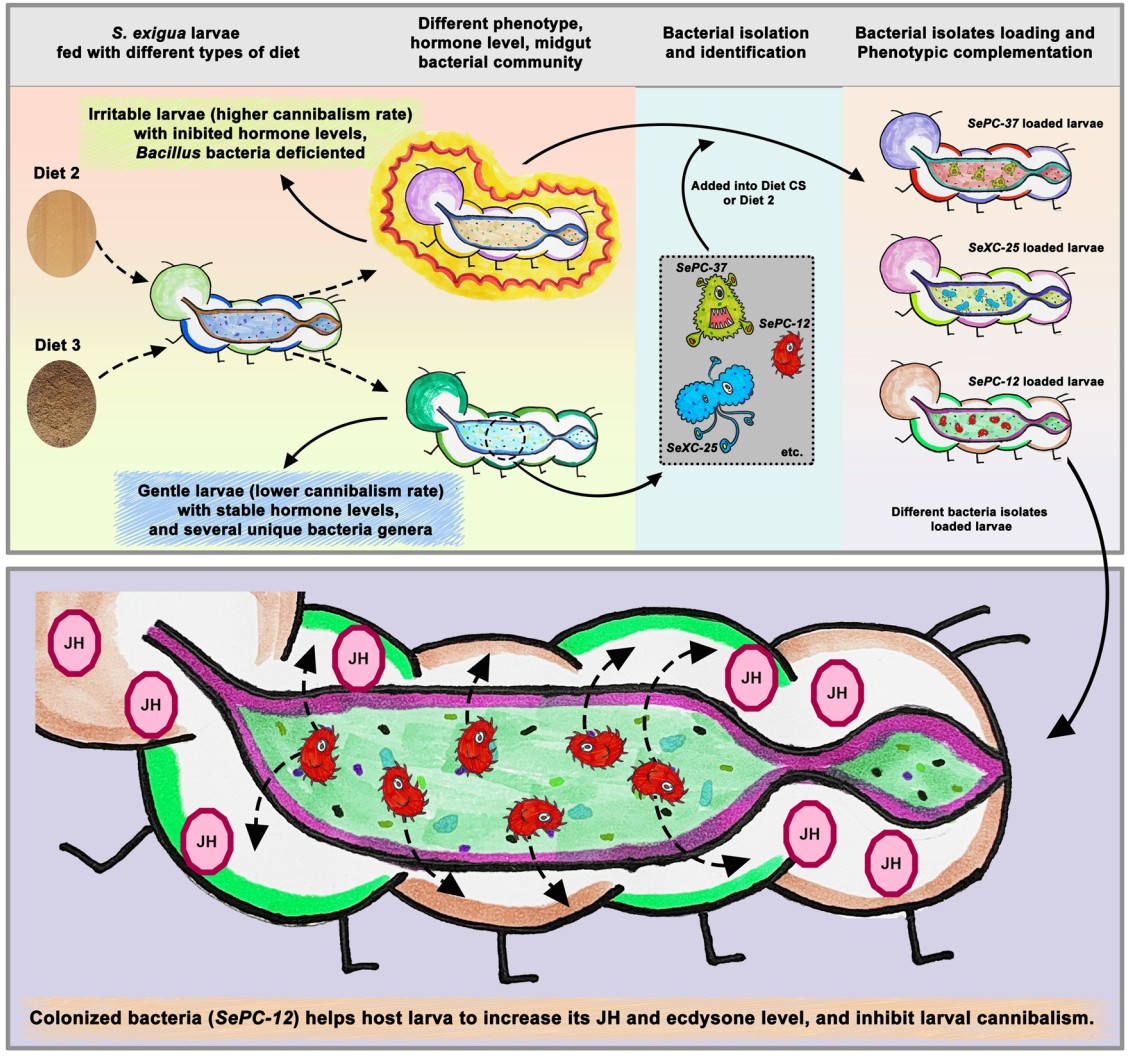

**Fig. 9 A proposed impact of gut bacteria on host larval cannibalism.** Feeding with different diets results in different larval midgut microbiota commits. Larvae fed with different diets had varied growth, development, or larval cannibalistic behavior. *SePC-12* isolated from the midgut of *S. exigua* larvae fed with the RW diet inhibited or delayed larval cannibalistic behavior after bacterial colonization. Furthermore, *SePC-12* loading increase larval juvenile hormone concentration, which was low in the aggressive larval group (CSK-fed *S. exigua* larvae). Therefore, *SePC-12* could help *S. exigua* balance the level of JH, inhibiting larval cannibalism.

the different feeding diets were transferred into bioassay chambers (hyaline cylindrical plastic canisters, $4 \times 6$ cm in height $\times$ diameter), in which a slice of the diet (approximately $2 \times 3 \times 0.3$ cm in width $\times$ length $\times$ height) was provided. Larvae fed a specific diet were used to determine cannibalism, and no cross-tests were performed. To test the larval cannibalism, 2, 5, 10, 15, and 20 larvae were put into the prepared bioassay chambers and reared in an incubator set at $27 \pm 1$ °C and $70 \pm 5\%$ RH. The residual diets were removed, and fresh diet dots were supplied daily. The larvae were observed daily. Larvae with missing body parts or apparent wounds (usually accompanied by gushed hemolymph or an extruded gut) were recorded as cannibalized. The number of cannibalized larvae was recorded daily, and the daily cannibalism rate was calculated. Six replicates were performed for each larval density of the five diets (Diets 1, 2, 3, 4, and 5). Heat maps of the average daily cannibalism rates of the larvae fed different diets were established using TB tools (GitHub, San Francisco, CA, USA)[71]. Data on daily larval cannibalism rates were analyzed using one-way ANOVA, and the least significant difference (LSD) was used to compare differences between the means of different dietary treatments using SPSS v22.0.

**Total bacterial load quantification and sex determination.** Twenty midguts of 3$^{rd}$ instar larvae maintained on five diets (Diets 1, 2, 3, 4, and 5) were dissected and weighed individually. To dissect the larval midguts, the insects were surface-washed with 75% ethanol for 90–100 s and rinsed thrice with double-distilled water. The midguts were dissected and pooled in a prechilled phosphate buffer solution (pH 7.4; 140 mM NaCl, 2.7 mmol/L KCl, 10 mM Na2HPO4, and 1.8 mmol/L KH2PO4). Thirty gut samples from identical diet-maintained larvae were pooled in Eppendorf tubes (Eppendorf, Hamburg, Germany). The total DNA of the gut was extracted using the SteadyPure Universal Genomic DNA Extraction Kit (Accurate Biology, Beijing, China), and the extracted DNA was used as a template for qPCR (SYBR® Green Premix Pro Taq HS qPCR Kit [Accurate Biology]). The total bacterial load was quantified using 16S rRNA universal primers (388F:5'-ACTCC-TACGGGAGGCAGCAG-3' and 806R:5'-GGACTACHVGGGTW TCTAAT-3')[72]. To determine the larval sex, two qPCR assays were performed to detect the expression level of the larval *kettin* and ATP synthase gene copy numbers (qkettin-F:5'-AGCACGATGT-GACGCCGAGTGT-3' and qkettin-R:5'-ATTGCTTGACCCT CGGTAACAA-3'; qATPs-F:5'-TCCTGCTGTTGTTCGCTTTC-3'

and qATPs-R:5′-CCACACATTCGATTCCATGGC-3′). The Kettin gene is a sex-linked gene without dosage compensation and is used to determine sex in different lepidopteran larval species[73,74]. Before qPCR, the PCR products of the bacterial 16S rRNA, larval kettin, and ATP synthase genes were purified and ligated into the pGEM-T Easy Vector (Promega, Madison, WI, USA) to generate standard molecules of specific genes. Standard curves for these genes were constructed by qPCR with a series of diluted plasmids. The absolute content of these genes was calculated using standard curves and was used for subsequent statistical analysis. The bacterial load in the gut of larvae fed different diets was calculated by dividing the bacterial 16S rRNA copy number by the gut weight. The differences in bacterial loads between larvae fed with different diets were analyzed using one-way ANOVA followed by LSD comparisons in SPSS v22.0 (IBM Inc). The ratio of the kettin copy number to the ATP synthase gene copy number was used to distinguish larval sex. A larva with a ratio of approximately 1 was deemed female, and a larva with a ratio of approximately 2 was considered male. The bacterial load between the male and female larvae within each diet-feeding group was compared using Student's $t$ test (SPSS v22.0 [IBM, Inc.]).

**Sample preparation, DNA extraction, and bioinformatic analysis.** The midguts of the 3rd instar *S. exigua* larvae fed Diets 1, 2, 3, 4, and 5 were dissected according to the abovementioned procedures. Thirty midguts (approximately 0.5 g) of each diet-maintained larva were collected in one tube and used as one biological replicate. Three replicates were used for each diet to maintain the larval samples. The dissected midguts were immediately stored in liquid nitrogen and sent to the Beijing Genomics Institute (Beijing, China) for DNA extraction, PCR amplification, sequencing, and bioinformatic analysis.

Briefly, DNA was extracted from dissected midgut samples using a FastDNA® SPIN Kit for Soil (MP Biomedicals, Solon, OH, USA). The V3–V4 hypervariable regions of the 16S rRNA gene were amplified by PCR using primer pairs 338F and 806R, as described above. PCR products were purified using Agencourt AMPure XP beads (Beckman Coulter Life Sciences, Lakeview, IN, USA), followed by tag addition and library construction. The size and concentration of fragments in the constructed libraries were determined using an Agilent 2100 Bioanalyzer (Santa Clara, CA, USA). Qualified libraries were sequenced on the HiSeq platform (Illumina) according to the size of the inserted fragments. The obtained reads were filtered, and the resulting clean reads were merged into tags using the Fast Length Adjustment of Short reads v1.2.11 (GitHub)[75], clustered into OTUs with a cutoff value of 97% using UPARSE software v7 .0.1090 (GitHub)[76], and chimera sequences were compared with the Gold database using UCHIME (v4.2.40) (GitHub)[77] for detection. OTU representative sequences were taxonomically classified using Ribosomal Database Project Classifier v.2.2 (Michigan State University, East Langsin, MI, USA) with a minimum confidence threshold of 0.6 and trained on the Greengenes database v201305 by QIIME v1.8.0 (GitHub)[78]. USEARCH_global (Github)[79] was used to compare all tags to OTU to obtain the OTU abundance statistical table for each sample.

The Venn plots of OTUs or taxa were plotted using the R package "VennDiagram" version 3.1.1 (University of Auckland, Auckland, New Zealand). Alpha and beta diversities were estimated using MOTHUR v1.31.2[80] and QIIME v1.8.0 (GitHub)[78] at the OTU level, respectively. Sample clustering was conducted using QIIME v1.8.0 (GitHub)[78] based on the Unweighted Pair Group Method with Arithmetic Mean (UPGMA). The Kyoto Encyclopedia of Genes and Genomes (COG) functions were predicted using the Phylogenetic

Investigation of Communities by the Reconstruction of Unobserved States[81]. Barplot and heatmap of different classification levels were plotted using R package v3.4.1 (University of Auckland) and R package "gplots," respectively. LEfSe cluster or linear discriminant analysis was conducted using LEfSe.

**Isolation of culturable bacteria.** To isolate gut bacteria, 20 randomly selected 3rd instar *S. exigua* larvae from a group fed with Diet 3 were starved for 24 h. The larvae were surface disinfected with 70% ethanol and dissected in 10 mM sterile phosphate-buffered saline (PBS). The midguts were then pooled into a 1.5 mL Eppendorf tube (Eppendorf) and homogenized with a pestle in 500 μL PBS. Subsequently, 10-fold dilutions were prepared with the homogenized products, and 100 μL of $10^{-2}$, $10^{-3}$, and $10^{-4}$ dilutions were spread on Luria–Bertani (LB) agar plates (Bio World Merchandizing Inc, Irving, TX, USA). The plates were incubated in an incubator at 30 °C, and the appearance of morphologically distinct colonies was observed daily for up to 48 h. Colonies were differentiated based on their size, color, and morphology. Furthermore, a single representative isolate of each morphotype was subcultured on new plates. Pure cultures of bacterial isolates were prepared and stored in 20% glycerol at -80 °C at the Hunan Agricultural University, College of Plant Protection, Changsha, China.

**Phenotypic and genotypic identification of isolates.** Macroscopic and microscopic characterization of the isolates was performed and evaluated according to Bergey's Manual of Systematic Bacteriology 1 and 2[82,83]. The colony morphology of the bacterial isolates plated on nutrient agar plates was observed under a stereomicroscope (Carl Zeiss, Jena, Germany). Cell shape and mobility of the bacterial isolates were determined using a light microscope (Carl Zeiss).

Staining, catalase reaction, Voges–Proskauer reaction (V–P reaction), starch hydrolysis, and indole properties of each bacterial isolate were determined according to previously described previously[84,85]. To determine the genotype of each bacterial isolate, genomic DNA was extracted using the SteadyPure Universal Genomic DNA Extraction Kit (Accurate Biology) according to the manufacturer's protocols. The 16S rRNA genes of the bacterial isolates were amplified using universal bacterial primers (UNI16S-F and UNI16S-R), and primer information is provided in Table 2. The PCR products were analyzed by agarose gel electrophoresis (Thomas Fisher Scientific, Waltham, MA, USA) and purified using an AidQuick Gel Extraction Kit (Aidlab, Beijing, China). The purified PCR products were ligated into the pGEM-T easy vector (Promega, Madison, WI, USA) and transformed into *E. coli* TG1. Five white-positive clones were selected for each bacterial strain and sent to Hunan Tsingke Biotechnology Co. Ltd. (Beijing, China) for sequencing.

The obtained sequences were subjected to BLAST searches using the National Center for Biotechnology Information (NCBI) GenBank database for similarity evaluation of sequences[86]. All sequences were deposited in GenBank. The sequences were assembled, edited using BioEdit (BioEdit, Manchester, UK), and aligned using ClustalW[87]. A phylogenetic tree was constructed using the Molecular Evolutionary Genetics Analysis 5.05 package (MEGA software solutions, Madhurawadha, India) and the neighbor-joining method[88] with Kimura's 2-parameter model[89], and bootstrap values were based on 500 replicates.

**Screening of bacterial isolates for the inhibition of larval cannibalism.** The bacterial clones were picked and transferred to LB medium, followed by culturing in a 30 °C shaker (200 rpm) for

48–72 h. Slices of Diet 1 or Diet 2 (approximately $2 \times 3 \times 0.3$ cm in width $\times$ length $\times$ height) were thoroughly immersed in the cultured bacterial fluid ($OD_{600} = 1.0 \pm 0.2$), and air-dried for 15–30 min. The treated diet slices were then transferred into 6 cm-diameter bioassay chambers, with one slice in each chamber. Newly molted 3rd instar Diet 1- or Diet 2-fed larvae were starved for 12 h and then transferred to a treated Diet 1 or Diet 2 dot-containing chamber. Larvae-fed sterilized LB medium-treated diet dots were used as controls. Ten larvae were placed in each chamber, and six replicates were performed. The larvae were maintained in an incubator set at $27 \pm 1$ °C and $70 \pm 5$% RH. The residual diets were removed and replaced with fresh diet dots containing bacterial fluid or LB medium daily until all tested larvae died or pupated. The larvae were observed daily. Larvae with missing body parts or wounds were recorded as cannibalized larvae. The number of cannibalized larvae was recorded daily, and the daily cannibalism rate was calculated. Heat maps of the average daily cannibalism rates of the larvae fed with different bacterial fluids were established using TBtools (GitHub)[71]. Data on daily larval cannibalism rates were analyzed using one-way ANOVA, and the least significant difference (LSD) was used to determine statistical differences between the means of different diet treatments in SPSS v22.0.

**Bacterial isolates loading confirmation**. To determine bacterial loading, the larvae maintained in Diets 1 and 2 were fed with diet dots dipped with *SePC-2*, *SePC-12*, *SePC-20*, *SePC-26*, *SeXC-27*, *SeXC-35*, or *SeXC-36* (which showed inhibitory effects on host larval cannibalism during the previous experiments), and the treated larvae were collected at 24 and 48 h post-treatment. The larval midguts were then dissected, weighed, and used to extract total genomic DNA, as described above. Bacterial numbers were determined using qPCR to detect bacterial loading.

**Determination of insect hormones**. To determine the possible relationship between larval cannibalism and JH or ecdysone levels, battles were performed between two larvae fed with diets 1, 2, and 3. Two 3rd-instar larvae (fed Diet 1, Diet 2, or Diet 3) were placed in a chamber (hyaline cylindrical plastic canisters, $2.7 \times 3$ cm in height $\times$ diameter) without feed, and cannibalism was monitored every 12 h. To distinguish larvae from different diet-fed groups, markers of different colors (red, blue, and black) were used. For each in-group battle pair (Diet 1 vs. Diet 1, Diet 2 vs. Diet 2, and Diet 3 vs. Diet 3) and cross-group battle pair (Diet 1 vs. Diet 2, Diet 1 vs. Diet 3, and Diet 2 vs. Diet 3), the time of occurrence of cannibalism and the winners were recorded. The winners were weighed, surface disinfected with 70% ethanol and stored at −80 °C for subsequent hormone determination. Each pair was replicated six times. Six 3rd instar larvae fed with Diets 1, 2, or 3 were starved for 24 h (used as CK) and subsequently used to determine basic hormone levels before cannibalism.

JH and ecdysone were extracted from winners and CK larvae according to previously described previously[90,91]. The Insect JH ELISA Kit (Cat#AJ-4940A, Changsha Aoji Biotechnology Co., Ltd., Beijing, China) and Insect Ecdysone ELISA Kit (Cat#AJ-3366A, Changsha Aoji Biotechnology Co., Ltd.) were used to determine the JH and ecdysone concentrations in each sample. Standard curves for JH and ecdysone are shown in Supplementary Fig. S6. JH and ecdysone concentrations in CK larvae or winners were analyzed using one-way analysis of variance (ANOVA), and the least significant difference (LSD) was used to compare means from different treatments using SPSS v22.0 (IBM, Inc).

**Functional confirmation of the isolated bacterial strains (*SePC-12* and *-37*)**. The functions of the two bacterial isolates (*SePC-12* and *-37*) were confirmed. First, the relationship between bacterial loading and larval hormone concentrations was determined. *SePC-12* and *-37* were smeared onto diet slices and fed to the 3rd instar *S. exigua* larvae fed with Diet 1 or Diet 2, as described above. The larvae were collected at 24 and 48 h post-treatment, and insect hormone concentrations were determined as described above. Larvae starved before bacterial feeding were used as controls.

To check whether the loading of these two bacterial isolates could inhibit the cannibalism of other noctuid larval species, *SePC-12* and *-37* were then used to feed Diet 1 or Diet 2 maintained *S. frugiperda*, *H. armigera*, and *H. assulta* larvae. Newly hatched *S. frugiperda*, *H. armigera*, and *H. assulta* larvae were fed with Diets 1 or 2 until they entered the 3rd instar stage. For each tested species, ten larvae were transferred into bioassay chambers with a consistent supply of diet slices smeared with LB medium, *SePC-12*, or *-37* culture mixtures. The larvae were maintained in an incubator set at $27 \pm 1$ °C and $70 \pm 5$% RH; the residual diets were removed and replaced with fresh diet dots daily. The number of cannibalized larvae was recorded daily, and the daily cannibalism rate was calculated. Heat maps of the average daily cannibalism rates of the larvae fed different bacterial fluids were established using TBtools (GitHub)[71]. The data on larval daily cannibalism rates were analyzed using one-way ANOVA, and the least significant difference (LSD) was used to determine statistical differences between means from different dietary treatments in SPSS v22.0 (IBM Inc).

**Generation of germ-free (GF) and mono-associated gnotobiotic *S. exigua* larvae**. To test whether *SePC-12* or *SePC-37* could grow under alkaline conditions, the pH of the LB culture medium was adjusted to 7.5, 8.0, 8.5, 9.0, 9.5, 10.0, 10.5, or 11.0. *SePC-12*, *SePC-37*, or *E. coli* TG1 were transferred into the above prepared different pH LB culture medium and cultured in a 30 °C shaker with a round per min of 200. Two hundred microliters of the bacterial culture mixture were divided into wells of a 96-well plate to measure the absorbance at optical density 600 ($OD_{600}$) every 4 h. The growth curves of *SePC-12*, *SePC-37*, and *E. coli* TG1 in LB culture media at different pH values were established using GraphPad Prism (v6.0, GraphPad Software).

The GF *S. exigua* larvae were then prepared according to the description of Chen et al.[92]. Briefly, the *S. exigua* eggs were surface sterilized by immersing in 1% NaClO (w/v) and 70% alcohol (v/v) before hatching, and then transferred into autoclaved Diet 3 contained in 10 cm Sterile Petri dish (Thermo Scientific). To confirm the GF condition, 30−50 mg of feces was collected from each Petri dish every 2 or 3 days. The feces samples were homogenized with 200 μL sterilized PBS solution, and 100 μL of them were spread on an LB agar plate and grew overnight at 37 °C. The larval-maintaining Petri dishes with bacterial colonies that appeared as plates were discarded and the larvae were confirmed under GF conditions and transferred into freshly prepared artificial diets containing Petri dishes to maintain a rearing density of no more than 40 2nd instar larvae per dish. Ten newly molted 3rd instar GF rearing larvae were collected to determine the gut bacterial load using qPCR according to the abovementioned procedures. According to the abovementioned procedures, another eight newly molted 3rd instar GF rearing larvae were collected to determine the JH concentration. The 3rd instar GF *S. exigua* larvae were then starved for 24 h (GF-SV) or supplied with a *SePC-12*-contained diet for 24 h, followed by starvation for another 24 h (SePC12-24). According to the procedures described above, the gut bacterial

load of GF-SV larvae or SePC12-24 larvae was determined by qPCR. The JH concentrations of the GF-SV larvae or SePC12-24 larvae were determined according to the abovementioned procedures. GF-SV larvae and SePC12-24 larvae were then used to perform in-group (IGGF or IGSePC12) and cross-group (GCGF/SePC12) battles, and JH concentrations were determined according to the procedures described above. The bacterial load between GF larvae or SePC12-24 larvae was compared using Student's *t* test (SPSS v22.0). The cannibalism starting times of the in-group and cross-group battles were analyzed using a one-way analysis of variance (ANOVA), and the least significant difference (LSD) was used to compare means from different treatments using SPSS v22.0. JH concentrations in different GF or mono-associated gnotobiotic larvae or winners were analyzed using one-way ANOVA, and the least significant difference (LSD) was used to compare means from different treatments using SPSS v22.0.

## Data availability

The datasets of all the produced reads for the bioinformatic analysis of Fig. 4 are available on GitHub (https://github.com/FiFiFishsh/For-mBio-20221105), and other datasets supporting the conclusions of this study are included in the article and its additional files. The source data behind the graphs in the figures are available in Supplementary Data 1.

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

## Acknowledgements
We thank the National Natural Science Foundation of China (32070168, 32172408) and Modern Agricultural Industrial Technology System of Hunan Province (2022-31, 2022-42) for their financial support.

## Author contributions
H.Y., A.-P.Z., and G.C. conceptualized and designed the study. X.-X.D., S.-K.C., H.-Y.X., and C.-J.Y. performed the experiments in the study. H.Y., A.-P.Z., and G.C. contributed reagents to the study. H.Y., X.-X.D., and C.-J.Y. analyzed the data. H.Y. and X.-X.D. wrote the manuscript. All authors read and approved the final manuscript.

## Competing interests
The authors declare no competing interests.
