## [Peer Review File · Communications Biology]

Reviewers' comments:

Reviewer #1 (Remarks to the Author):

The authors have done a nice piece of work related to the factors responsible for cannibalistic nature of a serious polyphagous pest. It is a very interesting study indicating the role of gut microbes in cannibalistic behavior of insects. These findings will be helpful in mass rearing of insects which show cannibalism. However, I have some queries as below:

1. The authors must mention that which generation of insects from the laboratory colony was used for experiments.
2. To study the culturable gut bacterial diversity, the larvae reared on RW diet were only used, secondly for screening of bacterial isolates for inhibition of cannibalism, CS and CSK diets were used. The basis for this should be mentioned.
3. What are CK larvae, should be given in M&M section.
4. For functional confirmation, 2 bacterial isolates i.e. SePC-12 and 37 were tested, what is basis of selection?
5. In M&M section line 515, bacterial isolates SePC-12 and -17 were tested for their functional role while in line 522, it is SePC 37, even in results Section it is SePC-37, may be typographical mistake. Need to be checked.
6. For testing the functional role of Bacteria SePC-12 and -37, in *S. frugiperda*, *H. armigera* and *H. assulta*, CS and CSK diets were used, while the culture of these insects were maintained on RW (*S. frugiperda* & *H. armigera*) and WS (*H. assulta*) diets. Why?
7. Nothing mentioned about the functional role of SePC-37 bacteria in results.
8. The experiments conducted on functional role of bacteria on other three insects should also be discussed. Please, add a short discussion about the possibility of using these bacterial isolates in diet for mass rearing.
9. Conclusion section also lacking significance of this study for future implementation.
10. Authority is not given with the scientific names of the insect. It should be written when the name comes first time in the text.

Reviewer #2 (Remarks to the Author):

It is interesting to study the potential relationship between insect cannibalism and gut bacteria, however I have several major concerns about the current work. A simple correlation might not be sufficient for reporting, the study of the behind molecular mechanism will improve this manuscript greatly.

It is well known that the lepidopteran gut is highly alkaline (pH >10); how can the *Escherichia* sp. (SePC-12) colonize the *Spodoptera exigua* larval gut and perform some functions? I suggest the authors to measure the gut pH in *S. exigua* larvae, and also test the growth of *Escherichia* sp. (SePC-12) under alkaline conditions.

Furthermore, the gut bacterial load (Fig. 3) looks like extremely high compared with other lepidopterans (please see: Caterpillars lack a resident gut microbiome). How did you quantify your samples? The qPCR method including the primer, cycling condition, melting curve, standard curve and so on need to be detailed. Since most of bacteria revealed here (Fig. 4) are easy to be cultured, the CFU analysis could also be performed to confirm the bacterial load.

It is unclear how the bacterial symbionts increase and stabilize larval JH concentration. How did you make sure that the larvae you employed for comparing the JH were at the same developmental stage? To clearly validate the role of gut microbiota in the host cannibalistic behavior, the authors are suggested generating germ-free/gnotobiotic *S. exigua* larvae and using them for subsequent cannibalism, determination of hormones, and other assays. One example, please check: Gut bacteria of the silkworm *Bombyx mori* facilitate host resistance against the toxic effects of organophosphate

insecticides.

The title can be revised, currently the statement ‘... inhibiting cannibalism under certain conditions...’ makes it sound uncertain.

The abbreviations used for different artificial diets in Abstract should be defined for clearance.

L122. The subtitle used here is not appropriate for the contents below as it provides an overview of the gut microbiota profiles observed across the various groups.

The authors need to justify why did they use the individuals fed on the RW-diet for bacterial isolation and not the other groups. Moreover, the subsequent smearing of the cultured bacterial isolates on the CS or CSK diet should be explained as well.

L337, 338. Here it says that the larvae were reared on the RW diet but L342 says that “Newly hatched *S. exigua* larvae were transferred to fresh diets and reared separately” This needs to be clarified.

L466-481. The bacterial concentrations used in the assay are unknown. Also, it is unclear for how long the larvae were provided with bacteria-amended diets.

L494-496. Please provide the catalog numbers of the kits.

L264, 265. ‘In the present study, sufficient food was provided; therefore, food quality influenced larval cannibalism’. Here, the authors established that ‘food quality influenced larval cannibalism’ but L267-269 somehow contradicts the claim. The tone needs to be neutralized.

L273-277. How come the apparent increase in the bacterial load of the baculovirus-infected larvae indicates that CSK-fed larvae may be more sensitive to entomopathogens?

L280-283. Provide references to the other studies.

L289-291. Based on the mentioned microbiota-mediated physiological processes, it is implausible to deduce that gut microbiota is associated with manipulating the host’s cannibalistic behavior. The authors should provide a more closely related example to make such an assertion.

L305-307. Only two studies do not qualify to be called “several reports Suggested....”

L251. lepidopterous, lepidopteran more often used.

L256. Should be ‘Except to the WY....’

Provide references L41-43, L251,

L274. This explanation is totally wrong, please read and cite the original literature carefully.

L65. ...or amino acids... remove the period.

L208. After exposure to cannibalism???

L329. Inactivation of what?

L333. Service life?

L337. N or D?

L519, 520. ‘Additionally, CS or CSK diets were fed hatched *S. frugiperda*, *H. armigera*, and *H. assulta* larvae until the 3rd instar stage’ What does it mean?

L285, 531... “OUTs” I think it’s OTUs.

The language need to be significantly improved.

Reviewer #3 (Remarks to the Author):

The manuscript titled “Feeding *Spodoptera exigua* (Lepidoptera: Noctuidae) larvae with bacteria increase larval juvenile hormone levels inhibiting cannibalism under certain conditions)” represent a significant contribution to scholarly research and is good for publication but with major revisions. There are lots of editorial issues in the manuscript which have been highlighted in the reviewed text that need to be addressed before acceptance for publication.

1. Abstract: The full meaning of the acronyms in the abstracts should be provided at the first appearance and subsequently used. No information on the bio informatics analysis done

3. In introduction: The introduction is very scanty, should be improved to provide the basic background information about the topic of study. Some parts are more like materials and methods eg- lines -50-51, 68-69 , while some are reflecting the results see lines-59- 61 - See more comments on the reviewed documents . Rewrite the introduction and state the clear objective of the study.

4. Materials and methods: This section has to be rearranged in chronological order as in the order they were done. See more comments on the reviewed documents. The arrangement is poor Materials

and methods needs major revision

5. Results. All the sub titles in the results section are not well captioned and should be rephrased- See my suggestion in the reviewed text.

The titles of all your figures should be rephrased and the foot note should be very short and clear – Unrelated figures should be separated

The manuscript needs a major revision , Other comments are found in the reviewed manuscript .

Response letter for COMMSBIO-23-0628

Reviewer: 1

1. The authors must mention that which generation of insects from the laboratory colony was used for experiments.

Response: We had state the information of the colonies of the insects as “The laboratory colonies of *S. exigua*, *S. frugiperda*, and *H. armigera* were maintained for more then 30 generations, and the field-collected *H. assulta* colony were maintained for more than 10 generations.” Please see the revised manuscript.

2. To study the culturable gut bacterial diversity, the larvae reared on RW diet were only used, secondly for screening of bacterial isolates for inhibition of cannibalism, CS and CSK diets were used. The basis for this should be mentioned.

Response: Many thanks. We had added explanation in the RESULTS, please see the revised manuscript.

3. What are CK larvae, should be given in M&M section.

Response: Sorry for our careless. The CK larvae were the starved larvae before they forced to battle. We had revised the M&M section. Please see the revised manuscript.

4. For functional confirmation, 2 bacterial isolates i.e. *SePC-12* and *-37* were tested, what is basis of selection?

Response: Many thanks. We had added explanation in the RESULTS, please see the revised manuscript.

5. In M&M section line 515, bacterial isolates *SePC-12* and *-17* were tested for their functional role while in line 522, it is *SePC-37*, even in results Section it is *SePC-37*, may be typographical mistake. Need to be checked.

Response: Sorry for our careless. The “*SePC-12* and *-17*” was corrected into “*SePC-12* and *-37*”. Please see the revised manuscript.

6. For testing the functional role of Bacteria *SePC-12* and *-37*, in *S. frugiperda*, *H. armigera* and *H. assulta*, CS and CSK diets were used, while the culture of these insects were maintained on RW (*S. frugiperda* & *H. armigera*) and WS (*H. assulta*) diets. Why?

Response: Sorry for our confusing description. We had added a sentence to simply explain the purpose of the experiment as “To check whether the loading of these two bacterial isolates can inhibit the cannibalism of other noctuid larval species, *SePC-12*

and -37 were then used to fed CS or CSK diets maintained *S. frugiperda*, *H. armigera*, and *H. assulta*.” Please see the revised manuscript.

7. Nothing mentioned about the functional role of *SePC-37* bacteria in results.

Response: Sorry for our careless. We had added the description of the function of *SePC-37* in the RESULTS. Please see the revised manuscript.

8. The experiments conducted on functional role of bacteria on other three insects should also be discussed. Please, add a short discussion about the possibility of using these bacterial isolates in diet for mass rearing.

Response: Thank you for your suggestion. We had added the discussion about the three noctuid larvae and the possible usage of the bacterial isolates in diet for mass rearing in the last two paragraphs in the DISCUSSION. Please see the revised manuscript.

9. Conclusion section also lacking significance of this study for future implementation.

Response: Thank you for your suggestion. We had added the significance of our study in the conclusion. Please see the revised manuscript.

10. Authority is not given with the scientific names of the insect. It should be written when the name comes first time in the text.

Response: Sorry for our careless. We had completed the authority of the scientific names of the insects. Please see the revised manuscript.

Reviewer: 2

1. It is well known that the lepidopteran gut is highly alkaline (pH >10); how can the *Escherichia sp.* (*SePC-12*) colonize the *Spodoptera exigua* larval gut and perform some functions? I suggest the authors to measure the gut pH in *S. exigua* larvae, and also test the growth of *Escherichia sp.* (*SePC-12*) under alkaline conditions.

Response: Thank you for your suggestions. We had tested the growth of *SePC-12* in different pH LB medium (pH ranged from 7.5-11.0), and the results showed that *SePC-12* grew well even in the pH 11.0 LB medium. Please see the added Figure 8 in the revised manuscript. Due to our lack of professional equipment, we did not measure the gut pH in different diets fed *S. exigua* larvae.

2. The gut bacterial load (Fig. 3) looks like extremely high compared with other lepidopterans (please see: Caterpillars lack a resident gut microbiome). How did you

quantify your samples? The qPCR method including the primer, cycling condition, melting curve, standard curve and so on need to be detailed. Since most of bacteria revealed here (Fig. 4) are easy to be cultured, the CFU analysis could also be performed to confirm the bacterial load.

Response: The sample preparing procedures were described in M&M: after the dissection, the gut samples were stored in 1.5 mL eppendorf tubes, one gut per tube. The gut samples were weighted by divided the weight of the empty tube (measured before the gut were dissected) from the gut contained tube. The total DNA of the gut samples were then extracted by using commercial kit as described in M&M. The PCR products were purified and ligated into pGEM-T easy vector to generate a standard vector for the following standard curve establish. The constructed V3-V4 fragment contained T vector were extracted and the plasmid DNA concentration were quantified by Nano Drop 2000 spectrophotometer (Thermo Scientific). The constructed vector were then diluted to prepare 5-fold series dilutions, and these dilutions were used as templates to perform qPCR with triple technical repeats. The qPCR protocol (A), melting curve (B), and standard curve (C) were provided as follow. As for the CFU analysis you suggested, we have not yet supplemented this part, which is mainly because the results of qPCR can already clearly compare the differences in the bacterial load in different treatments in our study.

3. It is unclear how the bacterial symbionts increase and stabilize larval JH concentration. How did you make sure that the larvae you employed for comparing the JH were at the same developmental stage? To clearly validate the role of gut microbiota in the host cannibalistic behavior, the authors are suggested generating germ-free/gnotobiotic *S. exigua* larvae and using them for subsequent cannibalism, determination of hormones, and other assays. One example, please check: Gut bacteria of the silkworm *Bombyx mori* facilitate host resistance against the toxic effects of organophosphate insecticides. The title can be revised, currently the statement ‘... inhibiting cannibalism under certain conditions...’ makes it sound uncertain.

Response: Thank you for your comments. As you say, it is unclear how the bacterial

symbionts increase the stabilize larval JH concentration, and this is what we will going to study in the next stage. The genomic DNA of *SePC-12* had just been sequenced, two gene was involved in catalytic conversion of farnesol into farnesic acid the insect hormone biosynthesis pathway. During the insect hormone determination in this study, we had tried our best to chose the larvae at the same developmental stage at the beginning of each experiments, and we also performed at least 6 biological repeats for each treatments. Our data more or less indicates that the some bacterial symbionts are indeed related to the larval hormone levels. Furthermore, we had performed the germ-free/gnotobiotic *S. exigua* larvae according to your suggestion, and the results suggested the conclusions that *SePC-12* gnotobiotic *S. exigua* larvae had increased JH concentration and was less aggressive. The title was revised into “Feeding *Spodoptera exigua* (Lepidoptera: Noctuidae) larvae with a *Escherichia sp. (SePC-12)* increase larval juvenile hormone levels inhibiting cannibalism”. Please see the revised manuscript.

4. The abbreviations used for different artificial diets in Abstract should be defined for clearance.

Response: Thank you for your suggestion. For the limit of the Abstract is 200 words, we had revised the abbreviations of different artificial diets the ABSTRACT. Please see the revised manuscript.

5. L122. The subtitle used here is not appropriate for the contents below as it provides an overview of the gut microbiota profiles observed across the various groups.

Response: Thank you for your comments. The subtitle was revised into “Larval gut bacterial analysis based on 16S rRNA sequence.” Please see the revised manuscript.

6. The authors need to justify why did they use the individuals fed on the RW-diet for bacterial isolation and not the other groups. Moreover, the subsequent smearing of the cultured bacterial isolates on the CS or CSK diet should be explained as well.

Response: Thank you for your suggestion. We had added explanation of why we used the Diet 3 (RW-diet) fed larvae to perform the bacterial isolation and why we perform the bacterial functional analysis on Diet 1 (CS) or Diet 2 (CSK) in the RESULTS. Please see the revised manuscript.

7. L337, 338. Here it says that the larvae were reared on the RW diet but L342 says that “Newly hatched *S. exigua* larvae were transferred to fresh diets and reared separately” This needs to be clarified.

Response: Sorry for our confusing description. For the daily rearing and population reproduction, the larvae were reared with RW diet, but for experiments, the larvae

were reared with different diets after they hatched from the eggs. We revised the manuscript to give more clear description. Please see the revised manuscript.

8. L466-481. The bacterial concentrations used in the assay are unknown. Also, it is unclear for how long the larvae were provided with bacteria-amended diets.

Response: We had added the bacterial concentrations by adding their OD₆₀₀ value ranges. And we had explained the providing of bacteria-amended diets as “The residual diets were removed and replaced with fresh diet dots with bacterial fluid or LB medium daily until all the tested larvae were dead or pupated.” Please see the revised manuscript.

9. L494-496. Please provide the catalog numbers of the kits.

Response: The catalog numbers had been provided, please see the revised manuscript.

10. L264, 265. ‘In the present study, sufficient food was provided; therefore, food quality influenced larval cannibalism’. Here, the authors established that ‘food quality influenced larval cannibalism’ but L267-269 somehow contradicts the claim. The tone needs to be neutralized.

Response: Thank you for your suggestion. The sentence was revised into “In the present study, sufficient food was provided; therefore, thus the larval cannibalism was mainly influenced by the food quality.” We hope the revised sentence would be better.

11. L273-277. How come the apparent increase in the bacterial load of the baculovirus-infected larvae indicates that CSK-fed larvae may be more sensitive to entomopathogens?

Response: Sorry for our inappropriate description. We had deleted the related sentences in DISCUSSION. Please see the revised manuscript.

12. L280-283. Provide references to the other studies.

Response: Sorry for our careless. The sentence was revised into “..... diet could influence the microbial gut composition of the *S. exigua* larvae, consistent with study reported by Gao et al.³⁶” Please see the revised manuscript.

13. L289-291. Based on the mentioned microbiota-mediated physiological processes, it is implausible to deduce that gut microbiota is associated with manipulating the host’s cannibalistic behavior. The authors should provide a more closely related example to make such an assertion.

Response: Sorry for our confusing description. The sentence was revised into “Gut

microbiota are reported to be involved in various physiological functions, including digestion and protection against pathogenic infections³⁸, and our study suggests that gut bacteria are associated with manipulating host cannibalistic behavior.” Please see the revised manuscript.

14. L305-307. Only two studies do not qualify to be called “several reports Suggested...”

Response: Sorry for our careless. The sentence was revised into “ It was reported to suggest that the larval cannibalistic behavior was correlated with JH concentration.....” Please see the revised manuscript.

15. L251. lepidopterous, lepidopteran more often used.

Response: Thank you, we had revised “lepidopterous” into “lepidopteran” according to your suggestion.

16. L251. L256. Should be ‘Except to the WY...’

Response: Thank you, we had revised “In addition to” into “Except to”.

17. Provide references L41-43, L251,

Response: Thank you, we had added several references. Please see the revised manuscript.

18. L274. This explanation is totally wrong, please read and cite the original literature carefully.

Response: Sorry for our careless, the sentence was revised by language editing company and we had not found that the meaning had changed. The sentence was revised into “Previous study has revealed that the bacterial load in the larvae gut is associated to the infection of baculovirus, for the gut microbiota can benefite the viral virulence, pathogenicity, and dispersion...” Please see the revised manuscript.

19. L65. ...or amino acids... remove the period.

Response: Sorry for our careless. The period was removed.

20. L208. After exposure to cannibalism???

Response: Sorry for our careless. The sentence was revised into “After cannibalism...” Please see the revised manuscript.

21. L329. Inactivation of what?

Response: Sorry for our careless. The sentence was revised into “The fermented

group A was then transferred into the autoclave at 121 °C for 20 min for sterilization.”

22. L333. Service life?

Response: Sorry for our careless. The sentence was revised into “To ensure the effectiveness of nutrients in the diet, the quality guarantee period of the prepared artificial diet did not exceed 10 days.”

23. L337. N or D?

Response: Sorry for our careless. The sentence was revised into “... a photoperiod of 16:8 h (L:D)...” Please see the revised manuscript.

24. L519, 520. ‘Additionally, CS or CSK diets were fed hatched *S. frugiperda*, *H. armigera*, and *H. assulta* larvae until the 3rd instar stage’ What does it mean?

Response: Sorry for our confusing description. The purpose of the designed experiments in the “Functional confirmation of the isolated bacterial strains (*SePC-12* and *SePC-37*)” part were briefly described. Please see the revised manuscript.

25. L285, 531... “OUTs” I think it’s OTUs.

Response: Sorry for our careless. The “OUTs” in the manuscript were corrected into “OTUs”. Please see the revised manuscript.

26. The language need to be significantly improved.

Response: Thank you for your comments. The language had been edited by Editage (www.editage.cn) before it was submitted, and we had sent it to Editage for the proofreading of the language again according to your suggestion. We hope the revised manuscript could meet your requirements.

Reviewer: 3

There are lots of editorial issues in the manuscript which have been highlighted in the reviewed text that need to be addressed before acceptance for publication.

Response: Thank you very much. We had revised the manuscript according to the attached file you sent to us. Please see the revised manuscript.

1. Abstract: The full meaning of the acronyms in the abstracts should be provided at the first appearance and subsequently used. No information on the bio informatics analysis done.

Response: Thank you for your comments. For the limit of the Abstract is 200 words, and the explaining the full meaning of the name that given to different artificial diet will use a lot of words. Besides, the acronyms of the diets doesn’t have any special

meaning. Thus, we had change the names of the six diet from Diet CS, CSK, RW, CSW, SW, and WY into Diet 1, 2, 3, 4, 5, and 6, respectively. We had added a sentence to describe the results of bioinformatics analysis according to your suggestion. Please see the revised manuscript.

2. In introduction: The introduction is very scanty, should be improved to provide the basic background information about the topic of study. Some parts are more like materials and methods eg. lines -50-51, 68-69 , while some are reflecting the results see lines-59- 61 - See more comments on the reviewed documents. Rewrite the introduction and state the clear objective of the study.

Response: Thank you for your suggestion. We had rewrite the INTRODUCTION. Some parts are the observed phenomenon during the decade of our insect rearing, and from the phenomenon we are interested in the influence of different diets on *S. exigua* larval cannibalism. So we did keep these sentence in the introduction. We hope the revised manuscript could meet your requirements.

3. Materials and methods: This section has to be rearranged in chronological order as in the order they were done. See more comments on the reviewed documents. The arrangement is poor Materials and methods needs major revision.

Response: Thank you for your suggestion. We had rearranged the order of the subsections in M&M according to your suggestion. Please see the revised manuscript.

4. Results. All the sub titles in the results section are not well captioned and should be rephrased- See my suggestion in the reviewed text.

Response: Thank you for your suggestion, we had revised all the subtitles in RESULTS according to your suggestion. Please see the revised manuscript.

5. The titles of all your figures should be rephrased and the foot note should be very short and clear–Unrelated figures should be separated.

Response: Thank you for your suggestion. We had revised figure legends, please see the revised manuscript.

Reviewers' comments:

Reviewer #1 (Remarks to the Author):

Many things are still not clear in the M&M section. Different diets were used for larval rearing for different experiments. The reason for this should be given in M&M section, instead of explaining in result section. The components of different diets given in Table 3 should come first i.e. Table 1 Discussion section not up to the mark. There is no need to repeat the results in discussion section, just discuss the findings giving reasons. Overall there are so many grammatical mistakes in the manuscripts.

Reviewer #2 (Remarks to the Author):

This revised manuscript has addressed most of my concerns, and overall, I believe it is worth publishing. A recent review (doi: 10.1146/annurev-ento-020723-102548), related with this direction, is recommended to be included in this manuscript. This review will strengthen the significance of the current work.

Response letter for COMMSBIO-23-0628

Reviewer: 1

1. Many things are still not clear in the M&M section. Different diets were used for larval rearing for different experiments. The reason for this should be given in M&M section, instead of explaining in result section. The components of different diets given in Table 3 should come first i.e. Table 1.

Response: Thank you for your comments. We had revised the manuscript. Since the order of the article is RESULTS before M&M, we have included some explanations in RESULTS instead of M&M. Besides, the explanation in M&M is also based on the obtained results, so we did not transfer the explanation in RESULTS to M&M. In addition, Table 3 has been moved to the first according to your suggestion. Please see the revised manuscript.

2. Discussion section not up to the mark. There is no need to repeat the results in discussion section, just discuss the findings giving reasons. Overall there are so many grammatical mistakes in the manuscripts.

Response: Thank you for your comments. We had revised the DISCUSSION according to your suggestion, and the language of the manuscript was revised by *EDITAGE*. The certifications of language editing were provided as follow. Please see the revised manuscript. We hope the revised manuscript could meet your requirements.

Reviewer: 2

1. This revised manuscript has addressed most of my concerns, and overall, I believe it is worth publishing. A recent review (doi: 10.1146/annurev-ento-020723-102548),

related with this direction, is recommended to be included in this manuscript. This review will strengthen the significance of the current work.

Response: Thank you for your suggestions. We had cited your suggested article as reference, please see the revised manuscript. We hope the revised manuscript could meet your requirements.